# Complete-Tree Space Favors Data-Efficient Link Prediction

**Chi Gao** [1]    **Lukai Li** [1]    **Yancheng Zhou** [2]    **Shangqi Guo** [1]

## Abstract

Link prediction is a fundamental problem for network-structured data. However, the prevalent research paradigm tends to assume abundant observed links, overlooking more challenging scenarios with a scarcity of observed links, which results in insufficient sample sizes. In real-world networks, hierarchical modularity, characterized by structured and nested connections, remains robust even with sparsely observed links. To address the challenge of limited link samples, we propose leveraging hierarchical modularity as a prior structure. We introduce complete-tree (CT) space, a discrete metric space with latent complete-tree structures, to formalize hierarchical modularity with an emphasis on its hierarchical permutation symmetry. Utilizing the group theory to quantize and compare permutation symmetries of different spaces, we prove that the CT space provides a significantly lower bound on sample complexity than the commonly used Euclidean space. We develop leaf matching, a data-efficient network embedding that maps nodes onto the CT space and conducts discrete optimization by reducing it to decentralized search. Experiments verify the data efficiency of CT space over other spaces. Moreover, leaf matching outperforms the state-of-the-art graph transformer in data-scarce scenarios while exhibiting excellent scalability. The code is available at: https://github.com/KevinGao7/LeafMatching.

## 1. Introduction

Link prediction is a long-standing and crucial problem in both graph representation learning and network science (Liben-Nowell & Kleinberg, 2003; Kumar et al., 2020).

[1]Center for Brain-Inspired Computing Research, Department of Precision Instrument, Tsinghua University, Beijing, China [2]Weixian College, Tsinghua University, Beijing, China. Correspondence to: Shangqi Guo <shangqi_guo@mail.tsinghua.edu.cn>.

*Proceedings of the 42nd International Conference on Machine Learning*, Vancouver, Canada. PMLR 267, 2025. Copyright 2025 by the author(s).

Its goal is to infer missing links in a network based on the observed ones, as real-world networks are often incomplete. Despite its significance, there exists a substantial gap between prevalent research settings and real-world demands: most models operate under the assumption that less than 15% of existing links are unobserved (Cen et al., 2023; Xiong et al., 2024), yet in numerous real-world networks, such as biological, informational, and social networks, the proportion of unobserved links can exceed by far that of observed links (Amaral, 2008; Olesen et al., 2011; Wei et al., 2016; Hohwald et al., 2009; Adafre & de Rijke, 2005). For instance, it is estimated that around 89% − 98% human binary protein interactomes are still not verified (Dunham & Ganapathiraju, 2021). Our work aims to fill this gap by investigating data-scarce regimes and tackling the challenge of insufficient sample sizes.

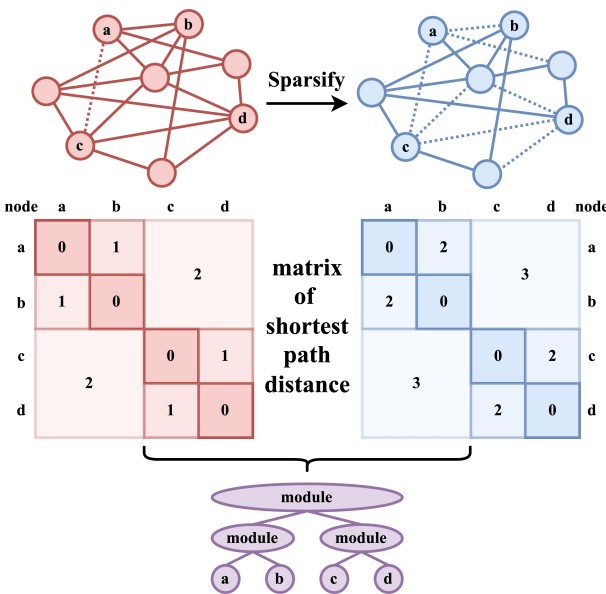

*Figure 1.* Hierarchical modularity exists regardless of link observability: the two distance matrices with different link observability both indicate that $a$ and $b$, as well as $c$ and $d$, each form a module, while the two formed modules together constitute a larger module.

Insufficient sample sizes call for strong priors, leading us to focus on hierarchical modularity, a general property in many important real-world networks, including but not lim-

ited to social, scientific collaboration, gene interaction, and brain networks (Chen & Li, 2015; Radicchi et al., 2004; Ryan et al., 2012; Meunier et al., 2010). In these networks, permuting nodes within modules of each layer preserves the overall distance matrix. Our key insight is that the hierarchically modular structure can still be detected even with low link observability, as shown in Figure 1. This property allows hierarchical modularity to serve as an effective structural prior, narrowing the hypothesis space of joint link distributions and reducing the sample complexity.

Hierarchical modularity inherently suggests hierarchical permutation symmetries in a network's latent space. For example, permuting nodes as $(a, b) \rightarrow (b, a)$, $(c, d) \rightarrow (d, c)$, $(a, b, c, d) \rightarrow (c, d, a, b)$ and their compositions does not alter the distance matrices in Figure 1. To formalize this, we introduce complete-tree (CT) space, a discrete metric space with latent complete-tree structures. In this space, the leaf nodes form the point set and the minimum height of common ancestor root nodes defines distances. The CT space fully captures the hierarchical permutation symmetry of hierarchical modularity in real-world networks, allowing it to intrinsically embed this prior and address the challenge of limited sample sizes. The upcoming challenge lies in the inherent discreteness of the CT space, which restricts the use of conventional mathematical tools commonly applied to continuous manifolds.

We solve this challenge by utilizing group theory. We define the *submetry group* by making an analogy with the classical concept of the isometry group, which captures the local permutation symmetries in a metric space. This group enables us to quantitatively calculate the hypothesize space size of joint link distributions. We can then derive a lower bound on the sample complexity for link prediction in any finite metric space of node representations. We prove that the CT space offers a significantly lower sample complexity than the commonly used Euclidean space while maintaining expressivity for networks with an arbitrary latent tree structure. Building on this, we develop leaf matching, a data-efficient network embedding that directly maps nodes onto the CT space. We further solve the discrete optimization challenge of leaf matching by converting it to an averaged version of greedy navigation over a generated computation graph that ensures scalable decentralized search.

Experiments among network embeddings verify the sample-efficiency of CT space compared to Euclidean, hyperbolic, and Hilbert space. Meanwhile, leaf matching outperforms the state-of-the-art graph transformer in data-scarce scenarios on real-world networks, while maintaining scalability. The contributions are listed below:

1. Theory. We provide a sample complexity lower bound for link prediction with nodes represented in an arbitrary finite metric space. We prove that the CT space offers a

significantly lower bound on sample complexity than the Euclidean space while maintaining expressivity for networks with an arbitrary latent tree structure.

2. Algorithm. We develop leaf matching, a data-efficient network embedding that maps nodes onto the discrete CT space, and construct its optimization algorithm.

3. Experiment. We verify the data efficiency of the CT space over Euclidean, hyperbolic, and Hilbert space and demonstrate the practicality and scalability of leaf matching.

## 2. Related Work

Despite the abundance of existing work on link prediction, both our problem focus and method innovations are unique.

### 2.1. Data-Efficient Graph Learning

Few-shot link prediction (Bose et al., 2019) aims to predict missing edges across multiple graphs using a small sample of known edges. This task targets applications where multiple graphs from a single domain are available. In contrast, our work focuses on a more challenging scenario where only one graph is known to the models. One-shot relation learning for knowledge graphs (Xiong et al., 2018) emphasizes leveraging the semantic knowledge extracted by the embedding models for relation generalization, while our method utilizes the general topological structure for data-efficient link prediction on complex networks. The work of (Kitsak et al., 2023) deals with the similar setting of large, substantially incomplete networks but targets at the shortest path node finding problem rather than link prediction.

### 2.2. Metric Space for Network Embeddings

The work of (Menand & Seshadhri, 2024) proves that low-dimensional vectors in Euclidean space cannot effectively capture the sparse ground truth of link distributions. In contrast, embeddings in hyperbolic space (Nickel & Kiela, 2017) demonstrate competitive performance regarding representation capacity and generalization ability for link prediction. Meanwhile, node2ket (Xiong et al., 2024), which operates in quantum Hilbert space, also exhibits excellent performance in both generalization capability and representation efficiency. Our CT space is comparable to hyperbolic space in terms of the exponentially expanded neighbors but differs due to its unique hierarchical permutation symmetries that theoretically ensures low sample complexity.

### 2.3. Tree-based Link Prediction

HBDM (Nakis et al., 2023) is a hierarchical block distance model that combines the clustered tree structure with Euclidean space vectors. The work of (Clauset et al., 2008) also employs the leaves of tree ensembles to represent each

node for link prediction. A fundamental difference between leaf matching and these existing works is that the complete-tree structure is pre-determined instead of data-driven in leaf matching. Therefore, no search is needed in selecting the optimal tree, and the hypothesis space size is greatly reduced to enable data-efficient link prediction.

# 3. Preliminaries

In this section, we will first formalize the link prediction problem in a supervised learning context and second describe the node representation framework on which our theory is built. We will then introduce the probably approximately correct (PAC) learning theory, which clarifies the main focus of our theory.

## 3.1. Link Prediction as Supervised Learning

Consider a graph $G = (V, E)$, with node set $V = \{1, ..., v\}$ and edge set $E \subset V \times V$. Formulating link prediction in a supervised learning context, we can see that the input set comprises node pairs $X = V \times V$, and the target set $Y = [0, 1]$ provides a continual quantification of link existence. Therefore, link prediction is exactly a regression problem. We restrict this study to undirected graphs.

The hypothesis space $H$ is then $h : X \to Y$. Let $L$ be a loss function, the generalization error of $h$ is thus $L_D(h) = \mathbb{E}_{x,y \sim \mathbb{D}(X,Y)}[l(h(x), y)]$, where $\mathbb{D}(X, Y)$ is unknown. The empirical error $L_S(h) = \frac{1}{s} \sum_{i=1}^{s} [l(h(x_i), y_i)]$, where $S = \{(x_i, y_i)\}_{i=1}^{s}$ is the training set for link prediction, comprising positive samples of observable links and generated negative samples.

## 3.2. Link Prediction in the Finite Metric Space

Consider a finite metric space $(M, d)$, with point set $M = \{1, ..., m\}$ of finite elements and the corresponding metric $d : M \times M \to \mathbb{R}$. Suppose the metric space has dimension $n$. In that case, we denote that an $n$ dimensional coordinate can represent each point in the point set, and that $m = m_1^n$, where $m, m_1$ respectively correspond to the number of points in the $n, 1$ dimensional metric space.

Our theory is built upon the node representation framework, where each node of a graph is represented by a point in the metric space (e.g., a coordinate vector in the Euclidean space), denoted by the mapping function $f : V \to M$. The link probability of each two nodes is negatively correlated with the distance of two node representations, denoted by the probability function $g : \mathbb{R} \to [0, 1]$, a strictly decreasing function with additional parameters related to the two nodes. Therefore, we can denote $h : X \to Y$ as a composition function: $h = g \circ d \circ (f \times f)$. This formulation is general for link prediction, regardless of whether an embedding or

neural network is used for deriving node representations.

## 3.3. PAC learning for Link Prediction

Since link prediction can be formalized as a supervised learning problem, the probably approximately correct (PAC) learning theory can be applied to calculate the sample complexity lower bound. Here we give the main result of PAC learning on which our theory is built (Shalev-Shwartz & Ben-David, 2014):

**Lemma 3.1.** *Suppose the hypothesis space $H$ is finite, the range of the loss function $l \in [0, 1]$, and that the training links $S$ are sampled i.i.d. from an arbitrary $D$, for any $\epsilon, \delta \in (0, 1)$, the sample complexity satisfies that if $s \geq \frac{\log(2|H|/\delta)}{2\epsilon^2}$, then with probability of at least $1 - \delta$, the generalization gap $|L_S(h) - L_D(h)| \leq \epsilon$ for all $h \in H$.*

By Lemma 3.1, a smaller hypothesis space size leads to lower sample complexity for a given generalization gap. Therefore, the key is to effectively reduce the size of the hypothesis space to constrain the sample error while maintaining expressiveness to ensure low approximation error.

# 4. Theoretical Bounds for the Complete-Tree Space and Others

In Section 4.1, we will first define the concept of the submetry group and, secondly, introduce the general sample complexity bound for an arbitrary finite metric space. In Section 4.2, we will formalize hierarchical modularity by a complete-tree-based metric space before calculating its sample complexity bound and comparing it with that of the Euclidean space. These two subsections focus on the sample error. In Section 4.3, we will discuss terms related to the approximation error by proving that the complete-tree space is sufficient to represent networks whose latent space has an arbitrary tree structure. All the proofs are in Appendix A.

## 4.1. Bound for Finite Metric Space

By Lemma 3.1, the key for estimating the sample complexity bound lies in calculating the size of the hypothesis space $H$. We can see from Figure 2 that the symmetry of the metric space directly constrains the hypothesis space size: the more symmetric the metric space is, the larger number of node placements will correspond to the same joint connection probabilities, and therefore the smaller hypothesize space size. Therefore, the challenge lies in giving a countable description of the symmetry of the metric space, which we solve by utilizing concepts in group theory, as shown below.

Since the nodes in $V$ will generally occupy only a subset of the whole point set $M$ instead of the whole set, the local symmetric properties of the metric space will more precisely constrain the hypothesis space size than the global ones.

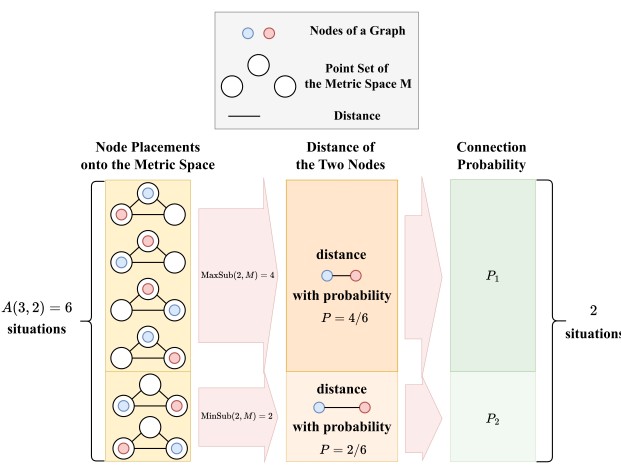

*Figure 2.* Since the distances of the three points in the metric space form an isosceles triangle, we can see that the local symmetry of the metric space constrains the size of the hypothesis space: the size is reduced to two despite the six situations of node placements.

However, no concepts in the mathematical literature, as far as we know, can directly describe such local symmetries. Therefore, before introducing the main theory, we will first define the submetry group by making an analogy with the existing concept of isometry group.

**Definition 4.1. Submetry Group**. The submetry group is defined to measure the local permutation symmetries of a finite metric space. Specifically, consider a subset of points $M_s \subset M$ in a metric space $(M, d)$ with $n$ dimensions, where $M_s$ satisfies that the coordinates of each point in $M_s$ do not overlap in each dimension. Denoting $\sigma$ as maps from $M_s$ to $M$, we define submetry group as the set of $\sigma$ that maintains all pair-wise distances:

$$\text{Sub}(M_s, M) \triangleq \{\sigma | d(i,j) = d(\sigma(i), \sigma(j)), \forall i, j \in M_s\}. \tag{1}$$

We denote the isometry group of $M$ as $\text{Iso}(M)$. Note that submetry group is a natural extension of the isometry group: when $M_s = M$ and $\sigma$ is restricted to a bijection, the submetry group becomes exactly the isometry group.

Different $M_s$ may have different orders of submetry groups, but we focus on boundary situations where the order of the submetry group is relevant only to the number of elements in the subset $M_s$. Thus, we define $\text{MinSub}(l, M) = \min_{|M_s|=l} (|\text{Sub}(M_s, M)|)$ to represent the minimal local symmetry of a metric space.

With the definitions above, we can now provide an exact estimation of the hypothesis size $|H|$ in the most general sense. Taking such an estimation into Lemma 3.1 of PAC learning, we can derive **the sample complexity lower bound for an**

**arbitrary finite metric space** as below.

**Theorem 4.2.** *Suppose the $v$ nodes of a graph are represented by $v$ points non-overlapping in each dimension of an $n$ dimensional finite metric space $M$ with $m$ points, where $m = m_1^n$. Suppose the link probability of each two nodes can be expressed by a strictly decreasing function with the distance of two node representations. Suppose the loss function $l \in [0,1]$, and the training links $S$ are sampled i.i.d. from an arbitrary $D$. For any $\epsilon, \delta \in (0,1)$, the sample complexity of link prediction satisfies that if*

$$s \geq s_{lb} = \frac{\log\left(2\left\lceil \frac{[A(m_1, v)]^n}{\text{MinSub}(v, M)} \right\rceil / \delta\right)}{2\epsilon^2}, \tag{2}$$

*then with probability of at least $1 - \delta$, the generalization gap $|L_S(h) - L_D(h)| \leq \epsilon$ for all $h \in H$. Here we denote $s_{lb}$ as the sample complexity lower bound, and $A(m_1, v)$ as the permutation number: choosing $v$ ordered elements from $m_1$ distinct elements.*

### 4.2. Bounds for the Complete-Tree and Euclidean Space

From Theorem 4.2, we can see that **the sample complexity lower bound for link prediction is directly constrained by the MinSub $(\mathbf{v}, \mathbf{M})$ of the metric space from which to represent the nodes.** We will start from the property of hierarchical modularity and see if it could lead to a metric space with a large order of the submetry group, and therefore constraining the sample complexity.

A module means that the nodes inside the module have a more significant connection probability than those outside. Therefore, module implies the symmetry of an arbitrary permutation of nodes inside the module. Hierarchical modularity thus implies that the latent metric space of the network has a hierarchical permutation symmetry, which is naturally satisfied by a tree. Accordingly, we formally define the complete-tree space below, visualized by Figure 3.

**Definition 4.3. Complete-Tree (CT) Space**. Consider a complete $b$-ary tree with $k$ layers. We define the distance between any two leaf nodes $i, j$ as $\frac{\text{lca}(i,j)}{k-1}$, where $\text{lca}(i,j)$ denotes the height of the lowest common ancestor node of $i, j$. Note that $\text{lca}(i,i) = 0$. Now consider the $n$-dimensional case: let the point set be the concatenation of leaf nodes: $M_{CT} = \{l : (l_1, l_2, ..., l_n)\}$ and metric $d_{CT}(l^{(a)}, l^{(b)}) = \frac{1}{n(k-1)} \sum_{i=1}^{n} \text{lca}(l_i^{(a)}, l_i^{(b)})$ for arbitrary $l^{(a)}, l^{(b)} \in M$. We can easily verify that $d_{CT} : M_{CT} \times M_{CT} \to \mathbb{R}$ is a metric and $d_{CT} \in [0,1]$. A CT space is exactly $(M_{CT}, d_{CT})$ with parameters $b, k, n$.

Given the definition of CT space, we can prove:

**Theorem 4.4.** *For a CT space with parameters $b, k, n$, suppose $l \leq b^{k-1} = m^{\frac{1}{n}}$, then $\text{MinSub}_{CT}(l, M_{CT}) > m^{\frac{l-1}{(k-1)(b-1)}}$.*

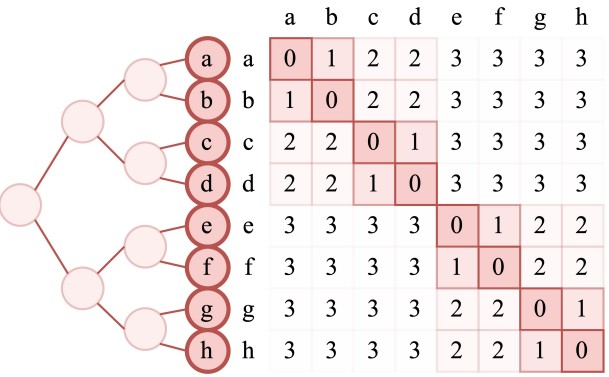

|   |   | a | b | c | d | e | f | g | h |
|---|---|---|---|---|---|---|---|---|---|
| a | a | 0 | 1 | 2 | 2 | 3 | 3 | 3 | 3 |
| b | b | 1 | 0 | 2 | 2 | 3 | 3 | 3 | 3 |
| c | c | 2 | 2 | 0 | 1 | 3 | 3 | 3 | 3 |
| d | d | 2 | 2 | 1 | 0 | 3 | 3 | 3 | 3 |
| e | e | 3 | 3 | 3 | 3 | 0 | 1 | 2 | 2 |
| f | f | 3 | 3 | 3 | 3 | 1 | 0 | 2 | 2 |
| g | g | 3 | 3 | 3 | 3 | 2 | 2 | 0 | 1 |
| h | h | 3 | 3 | 3 | 3 | 2 | 2 | 1 | 0 |

*Figure 3.* The complete-tree space has hierarchical permutation symmetries implied by the property of hierarchical modularity. For example, the permutation of $(a, b) \rightarrow (b, a), (a, b, c, d) \rightarrow (c, d, a, b), (a, b, c, d, e, f, g, h) \rightarrow (e, f, g, h, a, b, c, d)$ or their compositions does not change the distance matrix.

Now move on to the case of the Euclidean space, which is a default choice in graph representations. Consider an $n + 1$ dimensional Euclidean space with the commonly used cosine similarity as the metric. The point set in this space can be restricted to an $n$ dimensional sphere. When considering a discrete version of this space, reasonable due to the finite precision of real numbers in computers, such as float 64, there exist finite and countable points on the sphere. It is straightforward to prove that

**Theorem 4.5.** *For a discrete Euclidean space with cosine similarity as the distance,* $\mathrm{MinSub}_{Eu}(l, M_{Eu}) = 2m^{\frac{n+1}{2}}$, *where $n$ is the dimension of the sphere and $m$ is the number of points in the metric space.*

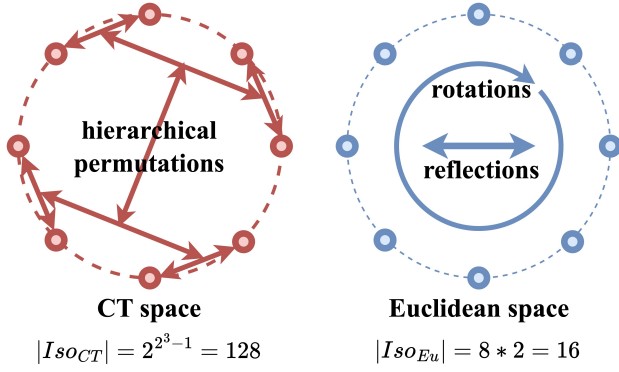

**CT space**
$|Iso_{CT}| = 2^{2^3 - 1} = 128$

**Euclidean space**
$|Iso_{Eu}| = 8 * 2 = 16$

*Figure 4.* This figure shows that the CT space has a much larger order of isometry group than the Euclidean space in the one-dimensional case. The comparison of submetry groups in multi-dimensional cases is more complex, but maintains the conclusion.

Comparing Theorem 4.4 with Theorem 4.5, and being aware that the number of nodes is generally much higher than the dimension of the metric space, we can see that

**Proposition 4.6.** *The Complete-Tree Space is more sample-efficient than the Euclidean Space for node presentation in link prediction.*

To understand intuitively why Proposition 4.6 holds, we can refer to the visual representation in Figure 4.

### 4.3. Expressivity of the Complete-Tree Space

While the previous subsections focus on the sample error, this subsection emphasizes the approximation error. Consider real-world networks with an arbitrary latent tree structure. Their latent space can be formalized by the *arbitrary-tree space*, which resembles the complete-tree space but allows for varying bifurcation numbers. In Theorem 4.7, we will demonstrate that a subspace within the complete-tree space can effectively approximate the arbitrary-tree space, as illustrated in Figure 5.

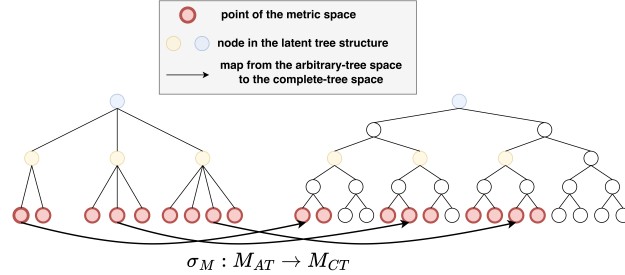

$$\sigma_M : M_{AT} \rightarrow M_{CT}$$

*Figure 5.* Subspace approximation: by increasing the number of layers, there exists a mapping $\sigma_M : M_{AT} \rightarrow M_{CT}$ from points in the arbitrary-tree space to the complete-tree space so that (1) the distance order is preserved; (2) the distance gap is constrained.

**Theorem 4.7.** *Subspace approximation. Consider an arbitrary-tree space with dimension $n$, with the tree structure in each dimension including $k_a$ layers. The bifurcation number to the $i$th layer ranges from $b_l^{(i)}$ to $b_h^{(i)}$, where $i = 0, 1, \ldots\ldots, k_a - 2$, and denote $r = \frac{\max_i(\lceil \log_b b_h^{(i)} \rceil)}{\min_i(\lceil \log_b b_h^{(i)} \rceil)}$, $r_1 = \max_i(\lceil \log_b b_h^{(i)} \rceil)$. Then there exists a map $\sigma_M : M_{AT} \rightarrow M_{CT}$ from the arbitrary-tree space to a complete-tree space with dimension $n$, bifurcation number $b$ and $k_c$ layers, where $k_c = 1 + \sum_{i=0}^{k_a-2} \lceil \log_b b_h^{(i)} \rceil$, so that (1) the distance order is loosely preserved; (2) the distance gap is constrained: $g = \frac{d_{CT}(\sigma_M(a), \sigma_M(b))}{d_{AT}(a, b)}$ satisfies that $g \in [\min(\frac{1}{r_1}, \frac{1}{r}), r]$. Loose order preservation: for an arbitrary three points $a, b, c$ in the arbitrary-tree space, if the distances in the arbitrary-tree space satisfies that*

$d_{AT}(a, b) < d_{AT}(a, c)$, *then in the complete-tree space, we have* $d_{CT}(\sigma_M(a), \sigma_M(b)) < d_{CT}(\sigma_M(a), \sigma_M(c))$.

Theorem 4.7 establishes that the complete-tree space is sufficient to represent a network with a latent space of arbitrary tree structures. This ensures that the approximation error will be low. **Note that this conclusion is general for real-world complex networks because hierarchy is a central organizing principle of complex networks as demonstrated in (Clauset et al., 2008).**

## 5. Leaf Matching

Leaf matching is an embedding that maps the nodes in graph $G$ to points in the CT space as defined in 4.3 with parameters $(b, k, n)$, denoted as $f : V \to M_{CT}$. We generally set $b = 2$ and $k$ for $b^{k-1}$ to approach the number of nodes in $G$.

### 5.1. Loss Function

The loss function of each node $i$ in graph $G$ is defined below:

$$\text{loss}(i) = -\left( \sum_{j \in P} \log(p(i, j)) + \lambda \sum_{k \in N} \log(1 - p(i, k)) \right), \tag{3}$$

where $P$ includes all the other endpoints of observable links that are connected to an endpoint of $i$. The negative nodes $k \in N$ are randomly selected $\deg(i)$ times from $V$ to serve as negative samples; $\lambda$ balances positive and negative losses. The probability $p(i, j)$ that each two nodes $i, j$ in graph $G$ has a link in the CT space is computed as below:

$$p(i, j; \alpha, \gamma) = \frac{1}{1 + \alpha \cdot \left( \frac{d_{CT}(f(i), f(j))}{\log [\deg(i) \deg(j)]} \right)^\gamma}, \tag{4}$$

where $\deg(i)$ denotes the degree of the $i$ th node, and $d_{CT}(f(i), f(j))$ denotes the distance of the two points in the CT space that represent nodes $i, j$. This formula is modified from (Krioukov et al., 2009; Jankowski et al., 2023), with an additional log term to match the exponentially growing neighborhood size of points in the CT space. Here $\gamma = 1, 2, 3, ...$ implies the clustering coefficient or inverse of temperature, which is unique to each type of complex network. $\alpha > 0$ is simply selected so that the loss is balanced between positive and negative samples, analytically calculable (Allard et al., 2024) when the network is scale-free.

### 5.2. Optimization Method

Since the CT Space is discrete, defining neighborhoods and gradients poses challenges. This limitation prevents the application of standard optimization methods used in continuous manifolds, such as stochastic gradient descent.

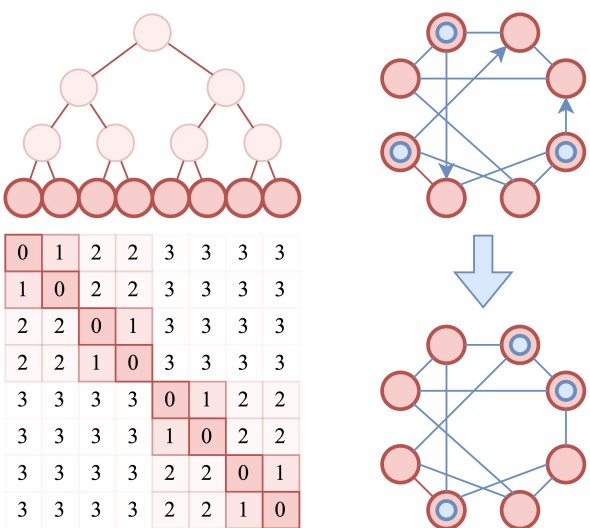

*Figure 6.* Leaf matching. The left side represents the latent tree structure and the distance matrix of the CT space. The right side shows the generated computation graph and the discrete optimization process. Specifically, the computation graph is derived from the CT space, and optimization occurs by moving to a neighbor node (arrowed) on the computation graph that has the lowest loss.

Consequently, we shift our focus to decentralized search techniques, which operate effectively in discrete domains and rely solely on local information.

Jon Kleinberg (2001; 2006) demonstrates that a specific subset of small-world networks can guarantee polylogarithmic time complexity for navigation between arbitrary pairs of nodes using a greedy strategy. This phenomenon suggests that the neighboring nodes in such networks can naturally define the neighborhoods within the discrete CT Space, facilitating scalable convergence. Building on this insight, the optimization algorithm for leaf matching is developed, reducing the network embedding optimization problem to an averaged version of greedy navigation.

Specifically, for each dimension of the complete-tree space, a computation graph $G_c$ is generated where the points (leaves) in the 1-dimensional CT space form the node set of $G_c$. Each node has a constant degree $k_d = c \log_b^2(n_l)$, where $n_l = b^{k-1}$ denotes the number of nodes in $G_c$ (selected to approach $|V|$ in $G$), and $c$ is a constant. The probability that arbitrary two nodes $x, y$ are connected in $G_c$ is given by the following expression

$$p_c(x, y) \propto \frac{1}{b^{d_{CT}(x,y)}}. \tag{5}$$

The optimization procedure is shown in Algorithm 1:

**Scalability.** The three loops in Algorithm 1 can all be

---

**Algorithm 1** Greedy-Navigation-based Optimization

---

**Input:** an graph to be embedded $G : (V, E)$ with $v$ nodes and $m_v$ edges; $d$ computation graphs $G_c : (V_c, E_c)$.

**Initiate:** Randomly map each node $i$ in $G$ to a node in each computation graphs $G_c$; the output is $f(i)$ with $d$ dimensions.

**repeat**

  **Add randomness.** If the current epoch is below a given ratio $r_e$ of the maximum epoch, randomly adjust all edges of a node to connect to a single random edge in the graph $G$ with a given probability.

  **for** $i = 1$ to $v$ **do**

    **for** $j = 1$ to $k_d$ **do**

      **for** $k = 1$ to $d$ **do**

        **Loss calculation.** Compute $\text{loss}(i)$ with the representation of $i$ in the $k$th computation graph now changed to be the $j$th neighbor node in $G_c$.

      **end for**

      **Greedy navigation.** Map the representation of $i$ in the $k$th computation graph to the neighbor node (including the current node) with the lowest loss.

    **end for**

  **end for**

**until** The training epoch reaches a given value.

---

parallelized through vector computations, resulting in a parallelizable time complexity of $O(m_v \log_b^2(n_l)d)$, where $d$ is the dimension of the CT space. It's important to note that we use $d$ here instead of $n$ to prevent any misunderstandings in the representation of time complexity; in other places of this paper, $n$ corresponds to the dimension.

Since integers can represent the leaf nodes, each point in the $d$ dimensional CT space can be represented by an integer vector. We generally choose $b = 2$ for leaf matching, so that both distance computations and edge samplings for the computation graphs can be accelerated through bit operations.

Since the greedy navigation for decentralized search has a guaranteed time complexity $O(\log_b(n_l))$ (Kleinberg, 2001), the number of convergence epochs for leaf matching is also $O(\log_b(n_l))$, which is demonstrated in the scalability experiment 6.3.2. Consequently, the total time complexity for leaf matching becomes $O(m_v \log_b^3(n_l)d)$. Note that $n_l$ is set to approach the number of nodes in $G$. Therefore, the discrete optimization algorithm is fast and scalable.

## 6. Experiment

We will outline experimental settings in Section 6.1, compare network embeddings across metric spaces to demonstrate the data efficiency of the CT space in Section 6.2, showcase the practicality and scalability of leaf matching in Section 6.3, and present ablation results in Section 6.4.

### 6.1. Settings

#### 6.1.1. DATASETS

We experiment on six datasets, including the classic Cora, Pubmed, Citeseer (Yang et al., 2016; Li et al., 2024), the large-scale ogbl-collab, ogbl-ppa (Hu et al., 2020), and the temporal ICEWS18 (Liu et al., 2022). These six typical benchmark datasets, which vary in size, include real-world citation, collaboration, political event, and protein interaction networks. Detailed data statistics are in Appendix B.

#### 6.1.2. EVALUATIONS

For Cora, Pubmed, Citeseer, and ICEWS18 we utilize *ROC-AUC*, *PR-AUC*, and *F1* score as the evaluation metrics. For the large-scale ogbl-collab and ogbl-ppa, we adopt the default evaluation metrics of *Hits@100* and *Hits@50*. We define **link observability** as the ratio $\mu$ between observable (training) edges and ground truth (total) edges. In the space comparison experiment, we vary $\mu$ from $0.9$ to $0.1$. In the practicality experiment, we set $\mu = 0.02$ to simulate practical scenarios. For data splitting and the negative ratio in testing, we adhere to the settings established by CogDL (Cen et al., 2023). Note that certain nodes from the test set are excluded if they lack links in the filtered training set.

#### 6.1.3. MODELS

There are twelve baselines. Three are representative network embeddings: node2vec (Grover & Leskovec, 2016), Poincaré (Nickel & Kiela, 2017), and node2ket (Xiong et al., 2024), respectively applying the Euclidean, hyperbolic, and quantum Hilbert space. Three are representative neural networks, including the state-of-the-art (SOTA) LPformer (Shomer et al., 2024) and two GNN baselines: SEAL (Zhang & Chen, 2018) and GAE (Kipf & Welling, 2016). The remaining six are heuristic algorithms: Adamic Adar (AA) (Adamic & Adar, 2003), Common Neighbors (CN) (Newman, 2001), Jaccard's Coefficient (JC) (Jaccard, 1901), Katz Index (KI) (Katz, 1953), Resource Allocation (RA) (Zhou et al., 2009), and Preferential Attachment (PA) (Barabási et al., 1999). All parameter settings are in Appendix B.

### 6.2. Data Efficiency of the Complete-Tree Space

In this experiment, we compare leaf matching using the complete-tree (CT) space with classic network embeddings from other spaces. Since all the models being compared are network embeddings, we anticipate that this experiment will effectively reveal the sample complexity of different spaces for node representation in link prediction. For fair comparisons, we set the dimension size to 16.

As shown in Figure 7, leaf matching generally outperforms all baselines across various link observability levels on representative datasets. Notably, as link observability $\mu$ decreases,

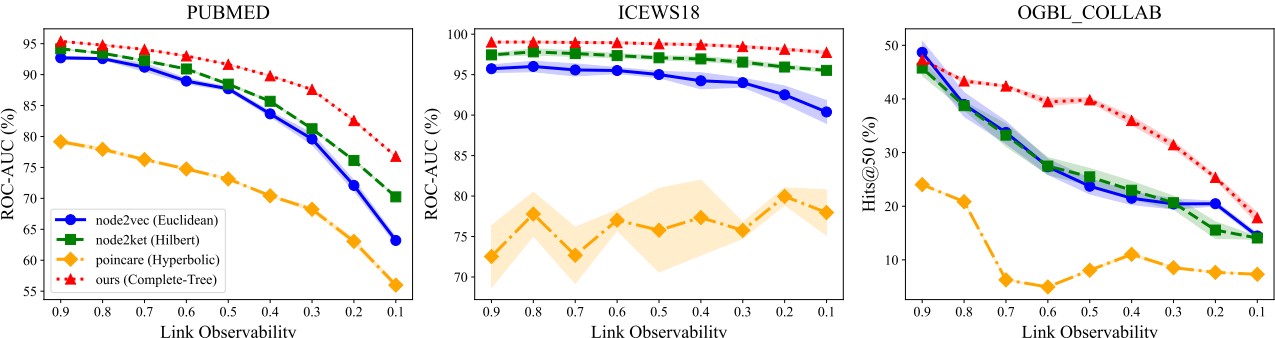

*Figure 7.* The performance of the four network embeddings across three representative datasets is presented, with link observability $\mu$ decreasing from 0.9 to 0.1. Experiments were repeated 5 times. The curves show the mean values, and the shaded areas represent the standard deviation. The full results (such as Cora, Citeseer and PR-AUC, F1 Score) can be found in Appendix B.

the advantage of leaf matching over node2vec increases, confirming the sample efficiency of the CT space compared to the Euclidean space, as established in Proposition 4.6.

Furthermore, the strong performance of leaf matching even at high link observability highlights the effectiveness of the CT space in representing complex networks, as demonstrated in Theorem 4.7. It is worth noting that the Poincaré embedding in hyperbolic space exhibits relatively low performance. We attribute this phenomenon to the fact that the Poincaré space has only one central point with exponentially expanded neighbors, which limits its effectiveness to tree-structured data. In contrast, the CT space allows every point to have exponentially expanded neighbors, enabling it to effectively represent real-world complex networks.

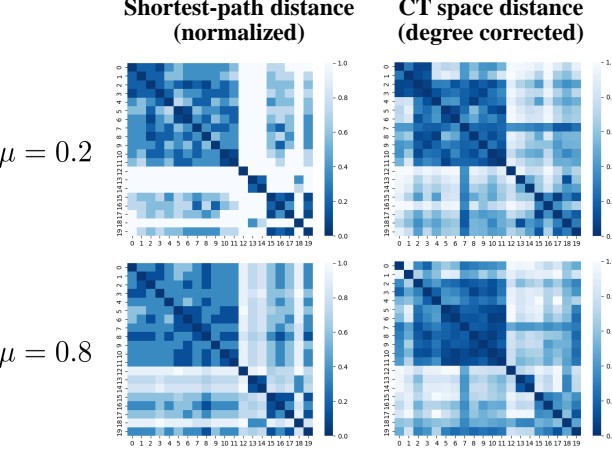

**Shortest-path distance (normalized)**     **CT space distance (degree corrected)**

$\mu = 0.2$

$\mu = 0.8$

*Figure 8.* Distance matrices of Citeseer are presented for nodes with the maximal degree. Additional visualization results can be found in Appendix B.

We can also see from Figure 8 that the hierarchically modu-

lar structures are generally detected and preserved by the CT space. At low link observability ($\mu = 0.2$), the structures are even sharpened, qualitatively demonstrating how and why the CT space is data-efficient.

### 6.3. Practicality and Scalability of Leaf Matching

#### 6.3.1. PRACTICALITY

*Table 1.* The link observability is set to $\mu = 0.02$ for simulating an extremely difficult setting with practical meanings, as around $89\% - 98\%$ human binary protein interactomes are still not verified (Dunham & Ganapathiraju, 2021). $Hit@50$ and $Hit@100$ are reported, respectively, for the ogbl-collab and ogbl-ppa datasets.

| MODEL | FEATURE | COLLAB(%) | PPA(%) |
|---|---|---|---|
| LPFORMER | ✓ | $15.55 \pm 1.63$ | $0.27 \pm 0.03$ |
| LPFORMER | ✗ | $8.43 \pm 3.05$ | $0.19 \pm 0.07$ |
| SEAL | ✓ | $5.26 \pm 1.46$ | OOM |
| SEAL | ✗ | $1.03 \pm 0.38$ | OOM |
| GAE | ✓ | $0.04 \pm 0.00$ | $0.00 \pm 0.00$ |
| GAE | ✗ | $0.21 \pm 0.10$ | $0.00 \pm 0.00$ |
| AA | ✗ | $3.52$ | $0.12$ |
| CN | ✗ | $3.52$ | $0.08$ |
| JC | ✗ | $3.52$ | $0.02$ |
| KI | ✗ | $6.76$ | $0.03$ |
| RA | ✗ | $3.52$ | $0.11$ |
| PA | ✗ | $2.02$ | $0.06$ |
| OURS | ✗ | $\mathbf{15.60 \pm 0.24}$ | $\mathbf{0.55 \pm 0.04}$ |

Table 1 shows that leaf matching outperforms all baselines, including the SOTA neural network (LPformer) even with the additional information on node features. Leaf matching is thus practical for link prediction in real-world low-data scenarios. Here we highlight some key applications where sparse link prediction is crucial: (1) scientific collaboration recommendation (Guo & Chen, 2013; Barabâsi et al., 2002);

(2) prediction of human protein interactomes (Dunham & Ganapathiraju, 2021); and (3) other important real-world networks such as mutualistic (Olesen et al., 2011), large social (Hohwald et al., 2009), and Wikipedia networks (Adafre & de Rijke, 2005), which have also demonstrated a higher prevalence of unobserved links compared to observed ones.

### 6.3.2. SCALABILITY

We conduct scaling tests using both synthetic (Erdős-Rényi random networks) and real-world (ogbl-collab) networks to verify the scalability of leaf matching.

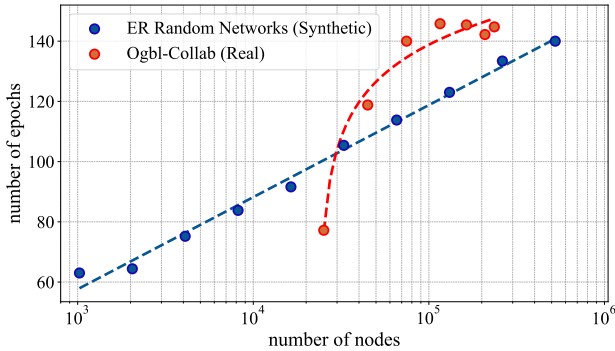

*Figure 9.* The convergence epoch is shown regarding the number of nodes in the networks.

Figure 9 shows that the convergence epoch of leaf matching is **linearly (or even sub-linearly) related to the logarithm** of the number of nodes in the graph, representing excellent scalability. This is because leaf matching is optimized on a specific computation graph guaranteeing polylogarithmic time complexity for navigation between arbitrary pairs of nodes using a greedy strategy, as described in Section 5.2.

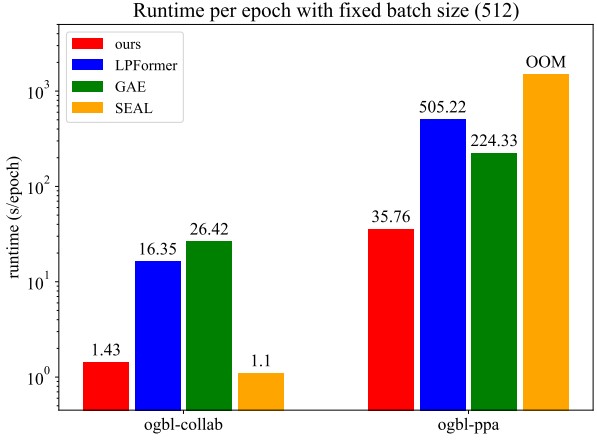

*Figure 10.* Runtime comparison between leaf matching and neural network baselines.

A complementary experiment comparing the detailed runtime per epoch is also conducted as shown in Figure 10, demonstrating the time efficiency of leaf matching. We attribute such a good property to the fact that each point in the $n$ dimensional CT space can be represented by an **integer vector**: we generally choose $b = 2$ for leaf matching, so that **bit operations** can be implemented to accelerate both distance computations in the CT space and edge samplings for the computation graphs.

### 6.4. Ablation Studies

The ablation experiments are conducted on Cora, Pubmed, and Citeseer, regarding $\alpha$ and $\gamma$ with $\mu = 0.1$. It can be seen from Figure 17 in Appendix B that leaf matching remains robust with varying hyperparameters.

## 7. Conclusion and Discussion

We systematically investigate data-efficient link prediction in metric spaces. We develop sample complexity theories from the perspective of permutation symmetry, introduce the data-efficient complete-tree space, construct a network embedding leaf matching with a scalable optimization algorithm, and validate our findings experimentally.

*Table 2.* Efficiency comparison between metric spaces. Specifically, data efficiency depends on the size of the hypothesis space, time efficiency depends on whether there exists a sub-quadratic optimization algorithm, and space efficiency depends on the typical vector dimension.

| Efficiency Type | Data | Time | Space |
|---|:---:|:---:|:---:|
| Complete-Tree Space | ✓ | ✓ | ✓ |
| Hilbert Space | | ✓ | ✓ |
| Hyperbolic Space | | | ✓ |
| Euclidean Space | | ✓ | |

We highlight above an efficiency comparison between commonly used metric spaces. It can be seen that the CT space is generally efficient. Furthermore, its ability to efficiently express arbitrary hierarchical structures suggests broader datatype extensions to enable data-efficient natural language processing (Tifrea et al., 2018), visual modeling (Zhuang et al., 2019), and hierarchical planning (LeCun, 2022).

In terms of insights for neuroscience, the CT space may also facilitate the functional understanding of hierarchically arranged grid cells in the entorhinal cortex (Zhang et al., 2023; Shpektor et al., 2024). Additionally, the proposed optimization method on the multi-dimensional CT space may reveal what the natural gradient in the brain is for, particularly in contexts where brain networks are greedily navigable (Seguin et al., 2018; Heszberger et al., 2021).

## Acknowledgements

We sincerely thank Luping Shi, Rong Zhao, Dahu Feng, Yihan Lin, Hao Zheng, Likai Tang, Sen Song, Huan Luo, Songhai Shi, and Ila Fiete for their valuable comments and encouragements. This work was supported by the National Natural Science Foundation of China (Grant 62206151) and China National Postdoctoral Program for Innovative Talents (Grant BX20220167).

## Impact Statement

This paper aims to advance the field of Representation Learning, with the positive social impact of facilitating a deeper understanding of critical networks, such as human protein interactomes, mammalian brain connectomes, and genetic interactions, even in the face of technological or funding constraints. However, there is a potential negative social impact when these techniques are applied to social networks, particularly regarding privacy concerns. It is essential to balance the benefits of enhanced understanding with the ethical implications of data usage in sensitive contexts.

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

# A. Theoretical Proofs

## A.1. Theorem 4.2

Since $M$ is a finite metric space, and that graph $G$ has finite nodes, we can see that the hypothesis space is finite. Therefore, the requirements of lemma 3.1 is satisfied and the sample complexity lower bound is determined by the size of the hypothesis space $|H|$.

Since the probability function $g : \mathbb{R} \rightarrow [0, 1]$ is strictly decreasing, $|H|$ equals the total number of the joint pair-wise distance distributions of the $v$ nodes in the metric space, which we denote as $|H| = |J|$, and $j \in J$ denotes a joint distance distribution.

By the assumption of non-overlapping mappings from $V$ to $M$, we know that there are totally $[A(m_1, v)]^n$ situations of node placements $F = \{f\}$. We can easily see from Figure 2 that there exists a mapping $K : F \rightarrow J$ that is a many-to-one surjective function.

For each $f \in F$, there is a $M_s(f)$, and therefore a $\mathrm{Sub}(M_s(f), M)$. By the definition of submetry group, we know that $\{\sigma \circ f | \sigma \in \mathrm{Sub}(M_s(f), M)\}$ is a set of node placements that belongs to the same joint distance distribution. We denote this set as $\mathrm{Set}(f)$. Meanwhile, for an arbitrary $f$, we have $|\mathrm{Set}(f)| \geq \mathrm{MinSub}(v, M)$. Therefore, an arbitrary distance distribution $j$ has at least $\mathrm{MinSub}(v, M)$ node placements.

Since $K : F \rightarrow J$ is a many-to-one surjective function, we can derive that $|H| = |J| \leq \lceil \frac{[A(m_1, v)]^n}{\mathrm{MinSub}(v, M)} \rceil$, and therefore theorem 4.2 by lemma 3.1.

## A.2. Theorem 4.4

Before bounding the $\mathrm{MinSub}(l, M_{CT})$ for the general $n$ dimensional CT space, we will first give the result of the more basic one-dimensional case.

For a CT space with parameters $b, k, 1$, suppose $v \leq b^{k-1}$ and denote the order of the isometry group of a one-dimensional CT Space as $I(b, k)$. Since $I(b, 1) = 1$ and $I(b, k + 1) = I(b, k)^b b!$ for $k = 1, 2, 3, ...$, we can derive $I(b, k) = b!^{\frac{b^k - 1}{b - 1}}$.

When it comes to the submetry group, its reduction of order from the isometry group happens when some leaves in a certain subtree are not fully occupied (permutations within these non-occupied leaves will not add to the order of submetry). Therefore, $\mathrm{MinSub}_{CT}(l, M_{CT}; 1)$ occurs when the leaves are occupied most intensively.

Accordingly, we can derive $\mathrm{MinSub}_{CT}(l, M_{CT}; 1) > I(b, 1 + \lfloor \log_b(l) \rfloor)$.

Now can we give the lower bound of $\mathrm{MinSub}_{CT}(l, M_{CT})$ for the more general $n$-dimensional case.

From the one-dimensional case, we know that $\mathrm{MinSub}(l, M_{CT}; 1)_{CT} > I(b, 1 + \lfloor \log_b(l) \rfloor)$, where $I(b, k) = b!^{\frac{b^k - 1}{b - 1}}$ for the 1 dimensional case. Therefore, $\mathrm{MinSub}(l, M_{CT}; n) > I(b, 1 + \lfloor \log_b(l) \rfloor)^n = (b!)^{\frac{b^{1 + \lceil \log_b(l) \rceil} - 1}{b - 1} n} > (b!)^{\frac{l - 1}{b - 1} n} = m^{\frac{l-1}{(k-1)(b-1)}} [(b-1)!]^{\frac{l-1}{b-1} n} \geq m^{\frac{l-1}{(k-1)(b-1)}}$.

## A.3. Theorem 4.5

The cosine similarity suggests that the $m$ points in the point set can be regarded as evenly distributed over a $n$ dimensional unit sphere, where the isometry group is composed of rotations and reflections.

By (Quessard et al., 2020), we know that rotations in the $n + 1$ dimensional space can be parameterized by the product of $\frac{(n+1)n}{2}$ unit rotations. Aware that there exists a one-to-one mapping between each rotation and each reflection (with a determinant of 1 and $-1$, respectively, in matrix representations), we can therefore derive that $|\mathrm{Iso}_{Eu}(M_{Eu})| = 2m^{\frac{1}{n} \frac{(n+1)n}{2}} = 2m^{\frac{n+1}{2}}$. Meanwhile, since the Euclidean space only has global symmetries and that the points in $M_s$ by definition can not by covered by an $n - 1$ dimensional sphere, we can see that $\mathrm{MinSub}_{Eu}(l, M_{CT}) = |\mathrm{Iso}_{Eu}(M_{Eu}) = 2m^{\frac{n+1}{2}}$.

## A.4. Proposition 4.6

From Theory 4.2, we know that the sample complexity lower bound is negatively correlated with $\mathrm{MinSub}(v, M)$. For the Euclidean Space with dimension $n$ (here $n$ refers to the dimension of the corresponding sphere), $\mathrm{MinSub}_{Eu}(v, M_{Eu}) = \mathrm{Iso}_{Eu}(M_{Eu}) = 2m^{\frac{n+1}{2}}$. For the CT space, $\mathrm{MinSub}_{CT}(v, M_{CT}) > m^{\frac{v-1}{(k-1)(b-1)}}$. Since the number of nodes in a graph is generally much larger than the dimension used, we can see that $\mathrm{MinSub}_{CT}(v, M_{CT}) > m^{\frac{v-1}{(k-1)(b-1)}} \gg \mathrm{MinSub}_{Eu}(v, M_{Eu}) = 2m^{\frac{n+1}{2}}$. Therefore, the CT Space is theoretically more sample-efficient than the Euclidean space for node representation in link prediction.

## A.5. Theorem 4.7

We first give the proof of the one-dimensional case.

Denote the coordinate of a point in the arbitrary-tree space as $x : (x_0, x_1, \ldots\ldots, x_{k_a-2})$, where $x_i$ denotes the serial number of the corresponding node in the bifurcation to the $i$th layer. For example, the point in the most left side of a tree has a coordinate of $(0, 0, \ldots\ldots, 0)$. Similarly, denote the coordinate of a point in the complete-tree space as $y : (y_0, y_1, \ldots\ldots, y_{k_c-2})$.

Construct the map $\sigma_M : M_{AT} \to M_{CT}$ such that $y = \sigma_M(x) = \mathrm{concat}_{i=0}^{k_a-2}(x_i)_{rb}$, where $(x_i)_{rb}$ denotes the reversed code of $x_i$ in the $b$-nary code with complementary zeros.

Consider two different points $a, b$ in the arbitrary-tree space, and denote $s$ as the location of the last different element of $x^{(a)}$ and $x^{(b)}$. We can see that $d_{AT}(a, b) = \frac{s+1}{k_a-1}$, where $s \in [0, k_a - 2]$. Meanwhile, we can derive that

$$d_{CT}(\sigma_M(a), \sigma_M(b)) = \begin{cases} \frac{k}{k_c-1} & \text{if } s = 0 \\ \frac{\sum_{i=0}^{s-1} \left\lceil \log_b b_h^{(i)} \right\rceil + k}{k_c-1} & \text{if } s > 0 \end{cases} \tag{6}$$

where $k \in [1, \left\lceil \log_b b_h^{(s)} \right\rceil]$ denotes the height of the lowest common ancestor node of the $s$th coordinate of point $a, b$, respectively, in a complete tree with bifurcation number $b$.

If $d_{AT}(a, b) < d_{AT}(a, c)$, the location $s(a, b) < s(a, c)$. Therefore, $d_{CT}(\sigma_M(a), \sigma_M(b)) < d_{CT}(\sigma_M(a), \sigma_M(c))$, the loose order preservation is proved.

Meanwhile, since $k_c = 1 + \sum_{i=0}^{k_a-2} \left\lceil \log_b b_h^{(i)} \right\rceil$, we can derive that

$$d_{CT}(\sigma_M(a), \sigma_M(b)) \in \begin{cases} \left[ \frac{1}{(k_a-1)r_1}, \frac{r}{k_a-1} \right] & \text{if } s = 0 \\ \left[ \frac{s}{(k_a-1)r}, \frac{(s+1)r}{k_a-1} \right] & \text{if } s > 0 \end{cases} \tag{7}$$

Therefore, we can derive that the distance ratio $g = \frac{d_{CT}(\sigma_M(a), \sigma_M(b))}{d_{AT}(a, b)}$ satisfies that $g \in [\min(\frac{1}{r_1}, \frac{1}{r}), r]$

For the $n$ dimensional case, we simply give the inequalities and gaps of distance components in each dimension, and since the average operation preserves the inequalities, the $n$ dimensional case is proved.

# B. Experimental Details

We first report some common settings of leaf matching in all experiments. Specifically, the bifurcation number of the complete tree is selected as $b = 2$. The epoch ratio to stop adding randomness is $0.8$. The probability of edge change in the randomness-adding process is $0.2$. Meanwhile, all the experiment results with variance are tested five times.

## B.1. Data Efficiency of the CT space

### B.1.1. PARAMETERS

We utilize the official implementation of node2vec, node2ket, and poincare to test their performance. Here's the hyperparameter settings for these algorithms in five datasets:

In general, we perform a grid search to systematically adjust the parameters of all models. The embedding dimensions are set to 16 for all models for fair comparison (For node2ket, it is the dimension of sub-embeddings that is set to 16). Other hyperparameters are set as default in their original official implementations except for the following adjustments for performance improvement:

For node2ket in Cora, Citeseer, Pubmed, and ICEWS18, we increase the number of sub-embeddings $C$ for each node from 8 to 16, the learning rate $\rho$ from 0.1 to 1.25 and decrease the number of iterations from 100M to 50M. For ogbl-collab, $C$ is increased to 32, $\rho$ is increased to 1.6 and the number of iterations is decreased to 50M.

For poincare in Cora, Citeseer, Pubmed, and ICEWS18, we increase the batch size from 5 to 64 and the learning rate from 0.001 to 0.01. For ogbl-collab, the batch size is further increased to 256 but the learning rate is set back to 0.001 for convergence of the loss.

For node2vec, no adjustments were made to the original parameters.

For leaf matching, in Cora, Citeseer and Pubmed, we set $h = 12, \gamma = 3, \alpha = 8, c = 1, \lambda = 1, r_e = 0.8$, training it with batch size 256 on 50 epochs. For ICEWS18, we set $h = 14, \gamma = 3, \alpha = 33, c = 3, \lambda = 1, r_e = 0.8$, training it with batch size 32 on 50 epochs. For ogbl-collab, we set $h = 16, \gamma = 3, \alpha = 57, c = 3, \lambda = 1, r_e = 0.8$, training it with batch size 256 on 50 epochs.

### B.1.2. DATASETS

We utilize the link prediction data wrapper from CogDL (Cen et al., 2023) to split the training and testing data. Here we show how the dataset statistics changes with the link observability $\mu$ for cora, pubmed, citeseer, ogbl-collab, and icews18:

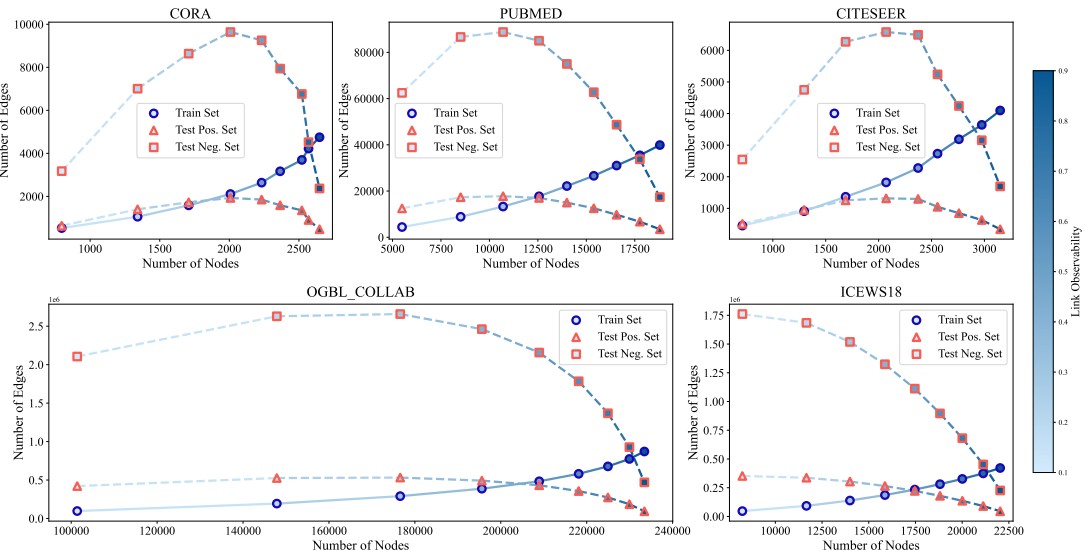

*Figure 11.* It is shown that the number of test edges has an unimodal distribution when link observability changes. This phenomenon is caused by the fact that some nodes are removed from the test set if they have no links in the filtered training set.

### B.1.3. EXTRA RESULTS

Reults on other metrics (ROC-AUC, PR-AUC, F1) for Cora, Citeseer, Pubmed and ICEWS18 are shown below in Figure 12, 13, 14, 15. The full visualization results are shown in Figure 16.

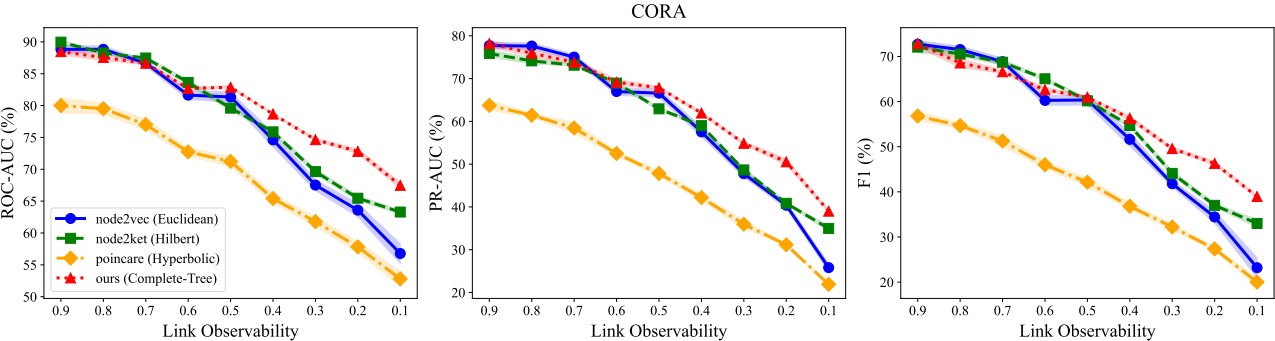

*Figure 12.* ROC-AUC, PR-AUC and F1 score of the network embeddings on Cora are presented, with link observability $\mu$ decreasing from 0.9 to 0.1. Experiments are run 5 times.

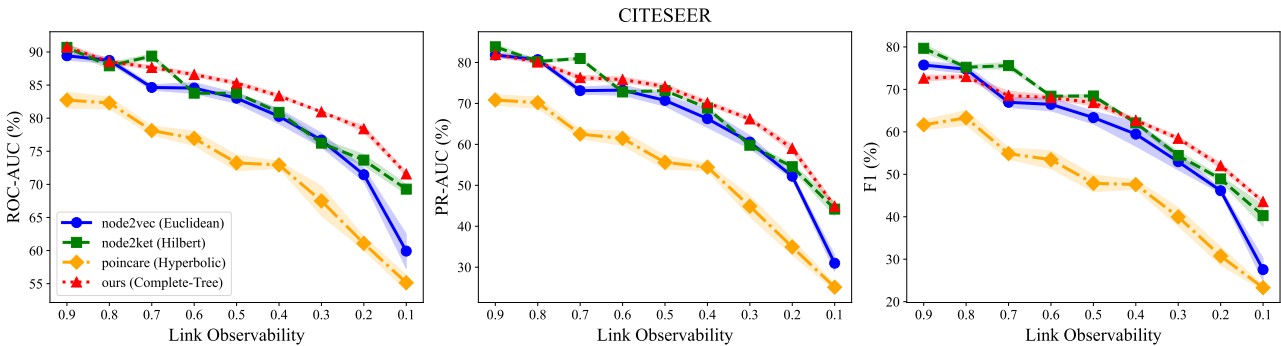

*Figure 13.* ROC-AUC, PR-AUC and F1 score of the network embeddings on Citeseer are presented, with link observability $\mu$ decreasing from 0.9 to 0.1. Experiments are run 5 times.

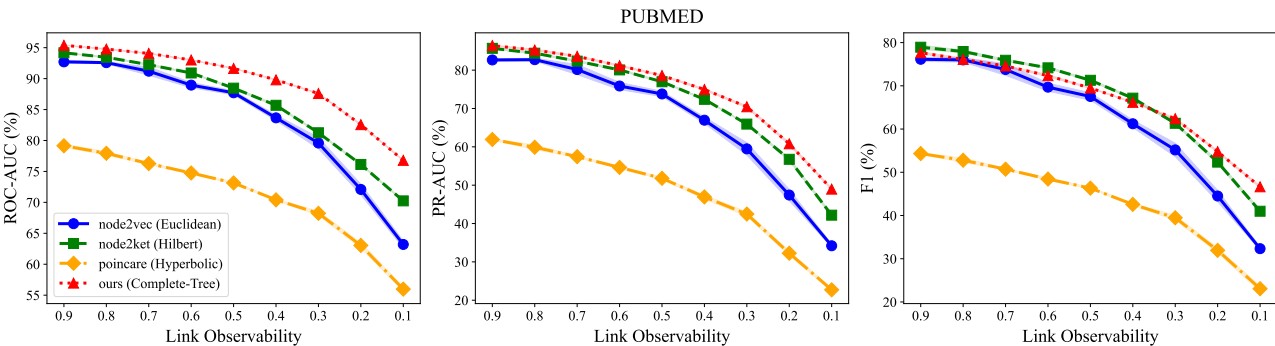

*Figure 14.* ROC-AUC, PR-AUC and F1 score of the network embeddings on Pubmed are presented, with link observability $\mu$ decreasing from 0.9 to 0.1. Experiments are run 5 times.

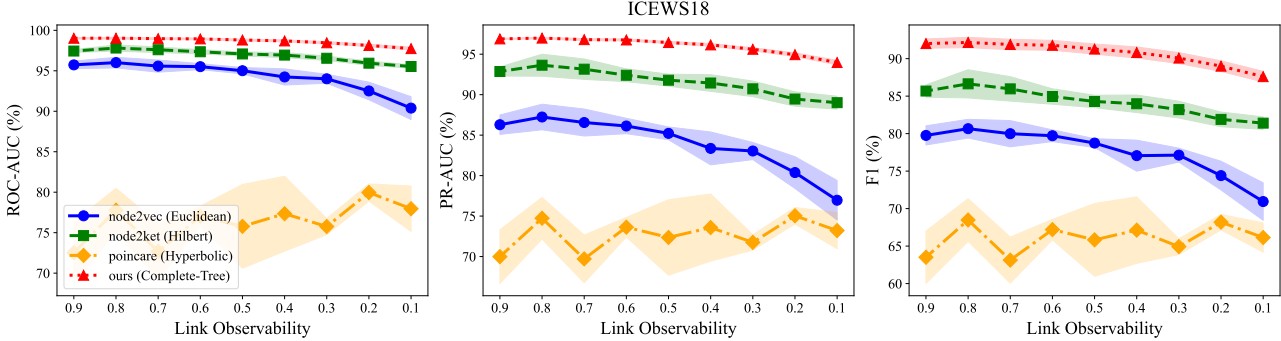

*Figure 15.* ROC-AUC, PR-AUC and F1 score of the network embeddings on ICEWS18 are presented, with link observability $\mu$ decreasing from 0.9 to 0.1. Experiments are run 3 times.

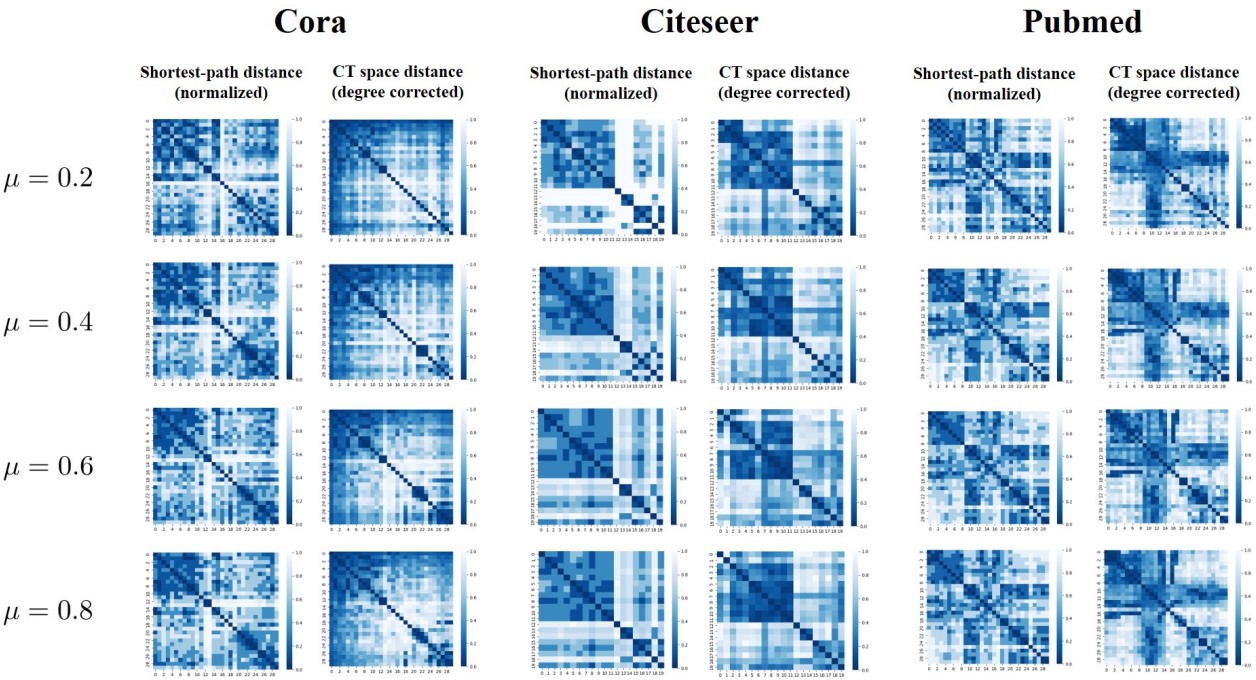

*Figure 16.* Distance matrices of Cora, Citeseer, and Pubmed are presented of nodes with the maximal degree, with link observability $\mu$ decreasing from 0.8 to 0.2.

## B.2. Practicality

### B.2.1. MODELS

We utilize the official implementations of LPFormer (Shomer et al., 2024), GAE (Kipf & Welling, 2016), and SEAL (Zhang & Chen, 2018) provided in HeaRT (Li et al., 2023) to evaluate performance on the ogbl-ppa and ogbl-collab datasets. The parameter settings for LPFormer are not changed. For SEAL, in addition to the original parameter settings, we set the number of training epochs to 100 and the evaluation step to 1. For GAE, since there are no official scripts for ogbl-collab and ogbl-ppa, we adopt the settings used for the PubMed dataset as a default. Similarly, we set the number of epochs to 100 and the evaluation step to 1.

For leaf matching, we set $h = 16, \gamma = 6, \alpha = 2.5, n = 16, c = 3, \lambda = \frac{4}{3}, r_e = 0.9$ in ogbl-collab. In ogbl-ppa, we set $h = 19, \gamma = 7, \alpha = 85, n = 32, c = 1, \lambda = 1, r_e = 1.0$. Both of them are trained with a batch size of 512, using 100 epochs for ogbl-collab and 20 epochs for ogbl-ppa.

Because there's no validation set in the setting of CogDL (Cen et al., 2023), we use the following policy to judge whether a neural network is converged. After applying Gaussian smoothing to the loss curve, we define the convergence point as the moment when the current loss $L_{\text{cur}}$ first satisfies:

$$\frac{1}{10}(L_{10\%} - L_{\text{cur}}) > L_{\text{cur}} - L_{\min} \tag{8}$$

In other words, we consider the neural network to have converged when the relative improvement in loss since the 10th percentile epoch exceeds ten times the gap between the current loss and the minimum loss.

Table 3. Heuristic methods for link prediction. $N(x)$ denotes the set of neighbors of node $x$. For the Katz index, $P(x, y, \ell)$ represents the set of all possible paths between nodes $x$ and $y$ of length $\ell$.

| Method | Scoring Function |
|---|---|
| Common Neighbors | $|N(x) \cap N(y)|$ |
| Preferential Attachment | $|N(x)| \cdot |N(y)|$ |
| Jaccard's Coefficient | $\frac{|N(x) \cap N(y)|}{|N(x) \cup N(y)|}$ |
| Adamic Adar | $\sum_{z \in N(x) \cap N(y)} \frac{1}{\log |N(z)|}$ |
| Resource Allocation | $\sum_{z \in N(x) \cap N(y)} \frac{1}{|N(z)|}$ |
| Katz index | $\sum_{\ell=1}^{\infty} \beta^\ell \cdot |P(x, y, \ell)|$ |

Heuristic methods predict missing links by first applying scoring functions $S(x, y)$ to pairs of nodes, and then comparing the scores of different links to make a judgment. As illustrated in Table 3, the six heuristic methods used in this study can be categorized based on the type of neighbor information they utilize.

1. **First-Order Method**: Common Neighbors (Newman, 2001), Preferential Attachment (Barabási et al., 1999), and Jaccard's Coefficient (Jaccard, 1901) rely solely on first-order neighbor information. These methods evaluate the immediate neighbors of nodes $x$ and $y$.

2. **Second-Order Method**: The Adamic Adar (Adamic & Adar, 2003) and Resource Allocation (Zhou et al., 2009) employ second-order neighbor information, calculating a weighted sum based on the properties of their common neighbors.

3. **Global Structure Method**: The Katz Index (Katz, 1953) calculates a weighted sum of all possible paths between two nodes, which can be exponentially numerous. To manage this complexity, we approximate the Katz Index by limiting the path length to 3 (essentially making it a 'third-order' method), represented as $\sum_{\ell=1}^{L} \beta^\ell \cdot |P(x, y, \ell)|$, where we set $\beta = 0.005$ and $L = 3$, consistent with the approach used in HeaRT (Li et al., 2023).

### B.2.2. DATASETS

For the two datasets ogbl-collab and ogbl-ppa, we utilize the link prediction data wrapper from CogDL (Cen et al., 2023) to split the training and test data. Besides the default setting, we set link observability $\mu = 0.02$. Note that certain nodes from the test set are excluded if they lack links in the filtered training set. Here's the information about the two datasets used in our setting:

*Table 4.* Dataset statistics for ogbl-ppa and ogbl-collab where the link observability $\mu = 0.02$ over the whole dataset. The number of negative edges is 5 times over the positive edges in the testing set as a default setting in CogDL.

| Statistic | ogbl-ppa | ogbl-collab |
|---|---|---|
| Original Nodes & Edges | 576,289 & 30,326,273 | 235,868 & 1,285,465 |
| Training Nodes & Edges | 328,160 & 424,639 | 31,107 & 19,353 |
| Testing Pos. Edges | 14,753,580 | 117,541 |
| Testing Neg. Edges | 73,767,900 | 587,705 |
| Training Edges / Testing Edges | 0.03 | 0.16 |

### B.3. Scalability

For synthetic data, the ER network is generated randomly with link probability $p = \frac{5}{|V|}$, where $|V|$ here denotes the number of nodes. We use the default setting of leaf matching, and adjust $k$ in each experiment to make $n_l = 2^{k-1}$ equals $|V|$. The convergence criterion is set to when the training loss is below $0.89$. Such a setting is reasonable because $0.89$ is generally the converging point of leaf matching over the ER networks of varying sizes. We repeatedly run the experiments five times and report the average converging epoch.

For real-world data, we partition the training edges in ogbl-collab into subsets containing $1, 1/2, ..., 1/64$ of the total edges. For each subset, we conduct five training runs, calculate the mean loss and smoothen the loss curve by the Gaussian Kernel. Convergence is determined when the relative decrease in loss for all subsequent epochs is no greater than $0.05\%$ compared to the current epoch. Additionally, the randomness technique used in neighbor selection is always kept (or $r_e = 1.0$) to consistently assess convergence. The fitting curve is log-log curve formulated by $y = \alpha \log \left( \beta \log \left( \gamma x + c \right) + b \right) + a$, where $x, y$ represent the number of nodes and the convergence epoch. Specifically, $\alpha = 20.97, \beta = 22.07, \gamma = 7.02 \times 10^{-4}, c = -5.54, b = -53.10, a = 62.04$.

### B.4. Run-time Comparison

We evaluate the per-epoch training time of our algorithm and other neural network models on the ogbl-collab and ogbl-ppa datasets where $\mu = 0.02$, with the batch size fixed at $512$. For SEAL on ogbl-collab, the reported runtime excludes the time required for subgraph dataset construction. On ogbl-ppa, SEAL encounters an out-of-memory (OOM) error, as constructing the subgraph dataset requires excessive memory (on the order of terabytes).

GAE is typically infeasible for training on the ogbl series datasets due to the need to reconstruct the entire adjacency matrix, which is prohibitive given the large number of nodes (over hundreds of thousands). To address this, we employ a block-wise reconstruction strategy for the adjacency matrix, enabling training without affecting the loss. However, this approach results in increased training time. The block size is set to $2048$, meaning that at each step, we reconstruct an adjacency matrix of size $(N, 2048)$ where $N$ is the number of nodes.

### B.5. Ablation Studies

It can be seen from Figure 17 that (1) leaf matching remains robust on the three datasets with varying hyperparameters; (2) space dimension has no significant influence on the performance of leaf matching, as long as it is not too small (above 8); (3) leaf matching achieves its best performance when the leaf number ($b^{k-1}$) approximates the number of nodes in the graph.

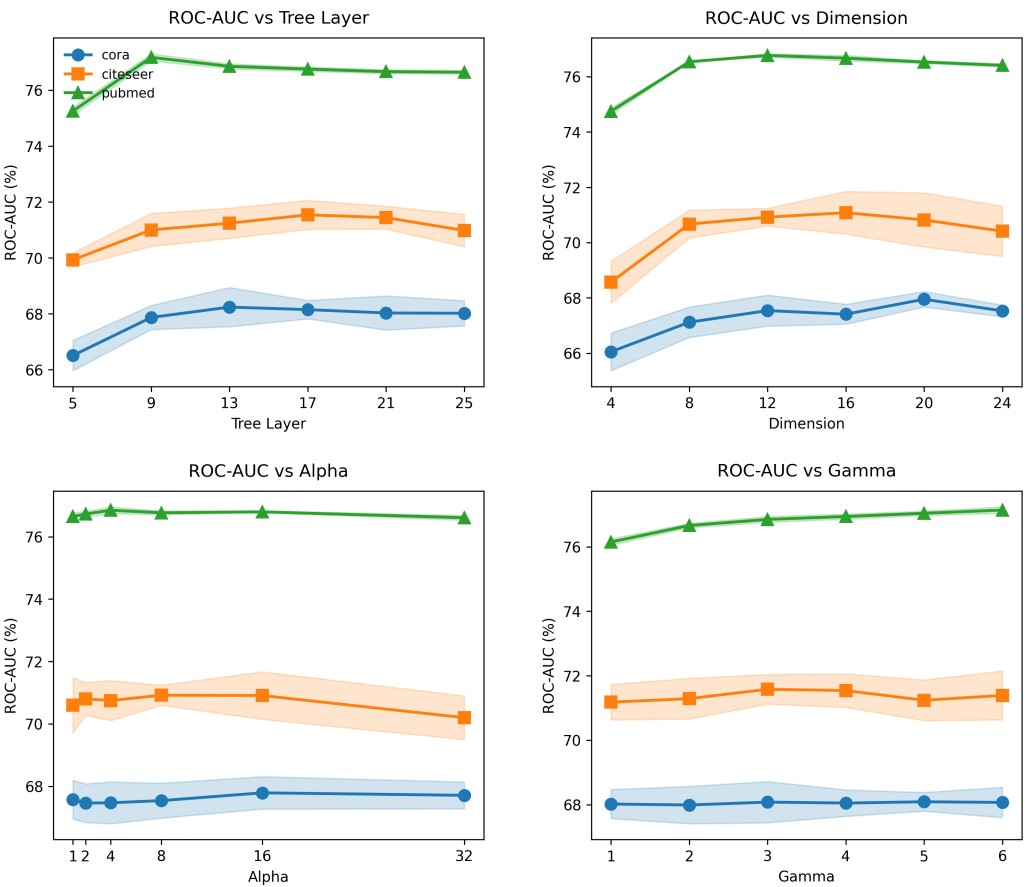

*Figure 17.* The ROC-AUC score with varying parameters on the three datasets.

