# OpenReview forum: "Complete-Tree Space Favors Data-Efficient Link Prediction"
_ICML.cc/2025/Conference — ICML 2025 poster_

### Official Review · Reviewer_yGLm · 2025-02-27

**Overall Recommendation:** 3

**Summary:**

This paper studies the link prediction task under the scarce data scenario. In the real-world application, the observed links in the network are far fewer compared to the unobserved ones. However, current LP methods usually focus on scenarios where observed links are abundant. To solve the problem, this study applies the hierarchically modular structure as the prior of the underlying graph structure through a "complete-tree" metrics space. A leaf matching method is proposed to map the nodes onto the complete-tree space. The experiments compared to other network embedding methods show that leaf matching in complete-tree space can achieve significant results when link observability levels are low.

**Claims And Evidence:**

- The claims made in the paper are mostly well supported by the theoretical analysis.
- However, the claim that "hierarchically modular structure can still be detected even with low link observability" in Introduction is not empirically validated. It will be more convincing if authors can empirically investigate how much of a hierarchically modular structure can be preserved in graph datasets from the experiment section, especially comparing the modular structure with different link observability $\mu$.

**Essential References Not Discussed:**

NA

**Experimental Designs Or Analyses:**

See "Claims And Evidence" part.

**Methods And Evaluation Criteria:**

- The method is well motivated to solve the problem at hand.
- However, the selection of benchmark datasets can be improved. When changing the link observability rate, the edges to drop are random, which does not reflect the growing trajectory of the graph. Authors can consider benchmarking the proposed leaf matching on graphs with temporal information on graph edges to better select the "observed" vs "unobserved" edges.
- More baselines should be considered in the region of low link observability. For example, heuristics link predictors like Common Neighbors, Resource Allocation, AdmicAdar Index, Katz Index, and Preferential Attachment should be included due to their strong baseline performance. GNN methods like GAE[1], SEAL[2] should also be discussed due to their wide adoption for LP tasks.

[1] Variational Graph Auto-Encoders
[2] Link Prediction Based on Graph Neural Networks

**Other Comments Or Suggestions:**

NA

**Other Strengths And Weaknesses:**

See above.

**Questions For Authors:**

- In experiments, why choose $\mu=$0.9 to 0.1 for Cora, Citeseer, Pubmed, but go aggressively on Collab and PPA with only $\mu=0.02$? Can authors show a similar results for OGB datasets like Figure 7 when varying $\mu$?

**Relation To Broader Scientific Literature:**

NA

**Theoretical Claims:**

Theoretical Claims is solid.

---

> ### Author Rebuttal · Authors · 2025-04-01
>
> We sincerely appreciate the reviewer for spending valuable time reviewing our manuscript and providing insightful comments. We have improved our paper accordingly and our responses are as below.
>
> Q1: **Hierarchical Modularity Detectability**: ‘the claim that "hierarchically modular structure can still be detected even with low link observability" in Introduction is not empirically validated…especially comparing the modular structure with different link observability $\mu$’
>
> A1: We thank the reviewer for highlighting this point and have clarified our claim in the revised paper. Theoretically, low link observability preserves hierarchical modularity because subsampling a stable distribution retains its statistical properties. To empirically validate this, we have added a **visualization experiment** for Cora, Citeseer, and Pubmed, as shown in https://anonymous.4open.science/r/Figures-58B7/Hierarchical%20Modularity.jpg. We can see that (1) hierarchical modularity persists even at low link observability; (2) leaf matching in the complete-tree space can well recover or even sharpen such a structure across the three datasets.
>
> Q2: **Temporally Growing Graph**: ‘Authors can consider benchmarking the proposed leaf matching on graphs with temporal information on graph edges to better select the "observed" vs "unobserved" edges’
>
> A2: This is a great suggestion. Since benchmark datasets for link prediction generally do not include elaborate temporal information, we have conducted **new experiments on the ICEWS18 dataset** from the domain of temporal knowledge graph, treating entities as nodes and relationships as edges. Results in https://anonymous.4open.science/r/Figures-58B7/Temporal%20Graph.png show that the complete-tree space significantly outperforms other spaces on this graph.
>
> Q3: ** Baselines**: ‘More baselines should be considered in the region of low link observability’
>
> A3: As suggested, we have added **6 heuristic and 4 GNN baselines** to the practicality experiment (see https://anonymous.4open.science/r/Figures-58B7/Practicality.png). Leaf matching remains competitive.
>
> Q4: **Link Observability Settings**: ‘In experiments, why choose$\mu=0.9$ to $0.1$ for Cora, Citeseer, Pubmed, but go aggressively on Collab and PPA with only $\mu=0.02$?’
>
> A4: We regard $\mu=0.02$ as an **extremely difficult setting with practical meanings**, as around $89 \\% - 98 \\%$ human binary protein interactomes are still not verified [1]. Higher $\mu$ naturally favors data-hungry neural networks over embedding methods. We emphasize that this work lays the foundation for future complete-tree-space neural networks, by theoretically, algorithmically, and experimentally demonstrating the data-efficiency of complete-tree space over other spaces.
>
> Q5: **OGBL for Space Comparison**: ‘Can authors show a similar results for OGB datasets like Figure 7 when varying $\mu$?’
>
> A5: We have **added ogbl-collab** to the space comparison experiments (see https://anonymous.4open.science/r/Figures-58B7/Ogbl-collab.jpg). We can see that the complete-tree space consistently excels.
>
> Once again, we would like to express our gratitude to the reviewer for the valuable feedback, which has helped us further improve our manuscript.
>
> Reference
>
> [1] Dunham, Brandan, and Madhavi K. Ganapathiraju. "Benchmark evaluation of protein–protein interaction prediction algorithms." Molecules 27.1 (2021): 41.

---

### Official Review · Reviewer_RBEo · 2025-03-09

**Overall Recommendation:** 4

**Summary:**

This paper addresses the understudied problem of link prediction when only sparse links are available. It proposes studying the complete tree space (CT), which offers a significantly lower bound on sample complexity compared to the commonly used Euclidean space. Building on this, the authors introduce leaf matching, a data-efficient network embedding that maps nodes onto the CT space and conducts discrete optimization by reducing it to decentralized search.

**Claims And Evidence:**

The claim that existing link prediction methods primarily focus on densely observed links and overlook cases with few links is valid. However, it would be beneficial to include practical use cases illustrating scenarios where sparse links are common.

**Essential References Not Discussed:**

N/A

**Experimental Designs Or Analyses:**

The experiments appear valid, but there is room for improvement:
1. The authors conduct CT space experiments on the first three datasets and leaf matching experiments on the last two, which seems like cherry-picking. It would be helpful if the authors could provide justification for this approach.
2. It would be beneficial to compare the proposed methods with traditional link prediction algorithms like AA, Common Neighbors, and Jaccard Similarity to justify their effectiveness.

**Methods And Evaluation Criteria:**

The proposed method involves CT space, which loosens the bound on sample complexity compared to commonly used metrics. The leaf matching algorithm effectively maps node embeddings to the CT space.

**Other Comments Or Suggestions:**

N/A

**Other Strengths And Weaknesses:**

The paper is well written and easy to read.

**Questions For Authors:**

It is necessary to justify the effectiveness of the proposed CT and leaf matching algorithms by comparing them with traditional link prediction algorithms. Many GNN-based link prediction works have been found to perform worse than traditional algorithms in link prediction. A comparison would strengthen the paper's claims.

**Relation To Broader Scientific Literature:**

N/A

**Theoretical Claims:**

I reviewed the proofs roughly and found them to be generally acceptable.

---

> ### Author Rebuttal · Authors · 2025-04-01
>
> We sincerely appreciate the reviewer for spending valuable time reviewing our manuscript and providing insightful comments. We have improved our paper accordingly and our responses are as below.
>
> Q1: **Practical Use Case**: ‘it would be beneficial to include practical use cases illustrating scenarios where sparse links are common’
>
> A1: Key applications where sparse link prediction is crucial: (1) **scientific collaboration recommendation** [1-2]; (2) prediction of human protein interactomes ($89 \\%-98 \\%$ unverified [3]); (3) other important real-world networks such as mutualistic networks [4], large social networks [5], and Wikipedia networks [6] have also demonstrated a higher prevalence of unobserved links compared to observed ones. We have integrated this discussion to the Practicality Experiment section.
>
> Q2: **Dataset Consistency**: ‘the authors conduct CT space experiments on the first three datasets and leaf matching experiments on the last two, which seems like cherry-picking. It would be helpful if the authors could provide justification for this approach’
>
> A2: We appreciate this important observation and have **added ogbl-collab and ICEWS18** (a temporally growing graph selected following the suggestion of Reviewer yGLm) to the space comparison experiment, which is shown in https://anonymous.4open.science/r/Figures-58B7/Ogbl-collab.jpg and https://anonymous.4open.science/r/Figures-58B7/Temporal%20Graph.png. It can be seen that the complete-tree space maintains more data-efficent than other spaces on both datasets. We conduct the practicality experiment only on the ogbl datasets because $\mu=0.02$ will lead to graph collapse on the three smaller datasets.
>
> Q3: **Traditional Baseline**: ‘It would be beneficial to compare the proposed methods with traditional link prediction algorithms like AA, Common Neighbors, and Jaccard Similarity to justify their effectiveness’
>
> A3: As suggested, we have added **6 heuristic and 4 GNN baselines** to the practicality experiment, as shown in https://anonymous.4open.science/r/Figures-58B7/Practicality.png. Leaf matching remains competitive.
>
> Once again, we would like to express our gratitude to the reviewer for the valuable feedback, which has helped us further improve our manuscript.
>
> Reference
>
> [1] Guo, Ying, and Xi Chen. "Cross-domain scientific collaborations prediction using citation." 2013 IEEE/ACM international conference on advances in social networks analysis and mining (ASONAM 2013). IEEE, 2013.
>
> [2] Barabâsi, Albert-Laszlo, et al. "Evolution of the social network of scientific collaborations." Physica A: Statistical mechanics and its applications 311.3-4 (2002): 590-614.
>
> [3] Dunham, Brandan, and Madhavi K. Ganapathiraju. "Benchmark evaluation of protein–protein interaction prediction algorithms." Molecules 27.1 (2021): 41.
>
> [4] Olesen, Jens M., et al. "Missing and forbidden links in mutualistic networks." Proceedings of the Royal Society B: Biological Sciences 278.1706 (2011): 725-732.
>
> [5] Hohwald, Heath, et al. "Inferring unobservable inter-community links in large social networks." 2009 International Conference on Computational Science and Engineering. Vol. 4. IEEE, 2009.
>
> [6] Adafre, Sisay Fissaha, and Maarten de Rijke. "Discovering missing links in Wikipedia." Proceedings of the 3rd international workshop on Link discovery. 2005.

---

### Official Review · Reviewer_fkGt · 2025-03-15

**Overall Recommendation:** 3

**Summary:**

This paper proposes a Complete-Tree based approach to explore the hierarchical structure of graphs such that the sample complexity for link prediction can be improved. The key idea is to develop a (hierarchical) complete tree to model the node distance in graphs, and utilize leaf matching to embed nodes into the tree, based on which the link probability can be computed. Experiments demonstrate the effectiveness and scalability.

**Claims And Evidence:**

Yes, claims are clearly stated with proofs given.

**Essential References Not Discussed:**

NA

**Experimental Designs Or Analyses:**

Yes. The method is tested on benchmark ogb datasets and the analyses are valid. However, more baseline methods (e.g., GNN-based methods) should be included, and more datasets (e.g., heterophilic graphs) should be considerd.

**Methods And Evaluation Criteria:**

Yes, the method explores the hierarchical structure in graphs to improve the scalability, and the evaluation criteria makes sense.

**Other Comments Or Suggestions:**

See questions

**Other Strengths And Weaknesses:**

(1)	The method is based on two assumptions: First, graphs exhibit hierarchical structures, and second, link probability is negatively related to the node representation distance. However, (First) though hierarchical structure clearly exhibit in synthetic graphs like CSBM, it is not clear how such ideal hierarchical structure exhibit in real-world graphs; (Second)the assumption on the relationship between link probability and node representation may not be applicable to heterophilic graphs.
(2)	The link prediction results are largely determined by the pre-defined features, e.g., CT, degree, node distances, etc. The proposed method may only be applicable to transductive settings but may not be applicable to inductive settings?
(3)	The proposed method can not incorporate node features into consideration.
(4)	Experiments: comparison with **more recent** GNN-based models should be included.

**Questions For Authors:**

(1)	Page 2 “Our key insight is that the hierarchically modular structure can still be detected even with low link observability, as shown in Figure 1”: Is there any empirical evidence to show the hierarchical structure exhibits in real-world graphs? Besides, the tree of real-world graphs may not be so ideal/balanced compared to synthetic graphs, will this affect the performance/scalability?
(2)	Section 4.1 Line 4-8: How do you measure the symmetry of the metric space? Can the proposed analysis generalizes to metric spaces that are not symmetric, e.g., directed graphs?
(3)	Def. 4.3: Is the complete tree spae determined once the graph is given? If a unique CT is not determineed, how to find the optimal tree? In general, the CT tree is somehow related to hierarchical graph clustering where graph clsuters (close nodes) at different levels are discovered. How do you compare the CT-based approach to other graph cluster approach? What are $l^{(a)}, l^{(b)}$ here?
(4)	Eq.(4): What’s the intuition? How to justify such formulation is (sub)optimal? How to choose alpha and gamma?
(5)	Experiments: more ablation/parameter studies should be included, e.g., effect of alpha, gamma; more baselines; more datasets.
(6)	Fig. 8: Maybe better to have x-axis as log10(number of nodes)
(7)	The template for this submission seems to be incorrect, as there’s no line number to refer to.

**Relation To Broader Scientific Literature:**

Yes

**Theoretical Claims:**

I checked the theorems in the paper but didn’t not review the details of the proofs.

---

> ### Author Rebuttal · Authors · 2025-04-01
>
> We sincerely appreciate the reviewer’s time and insightful comments. Our justifications and improvements are as below.
>
> W1: **Hierarchical Evidence**: **Numerous existing studies have empirically verified** hierarchical modularity in many important real-world networks: metabolic [1], customer purchasing [2], genetic regulatory or interaction [3-4], social [5], scientific collaboration [6], anatomical or functional brain networks [7], the Internet [8], etc. We have also added a **visualization experiment** to demonstrate the hierarchical modularity in three real-world datasets (see https://anonymous.4open.science/r/Figures-58B7/Hierarchical%20Modularity.jpg).
>
> W2: **Real-World Performance/Scalability**: Leaf matching has **excellent performance over real-world networks**. This is theoretically validated in Theory 4.7 and empirically verified in Sec 6.2-6.3. Briefly speaking, a subspace in the **complete-tree space** can well approximate the **arbitrary-tree space** and therefore guarantees good generalization. For further demonstration, we have added **ogbl-collab and ICEWS18** (a temporally growing graph selected following the suggestion of Reviewer yGLm) to the space comparison experiments (see https://anonymous.4open.science/r/Figures-58B7/Ogbl-collab.jpg and https://anonymous.4open.science/r/Figures-58B7/Temporal%20Graph.png). It is shown that the complete-tree space remains more data-efficient than other spaces on the million-scale real-world datasets. We have also conducted a **new scaling experiment** on the real-world ogbl-collab dataset using the $\log{10}$ scale, as shown in https://anonymous.4open.science/r/Figures-58B7/Scalability.png, representing even better scaling than on synthesized graphs.
>
> W3: **Heterophilic/Inductive/Features**: Our main focus is demonstrating **complete-tree space superiority over other spaces** for data-efficient link prediction, which we have verified theoretically, algorithmically, and experimentally. For these extended capabilities, the Euclidean-space adaptations [9-11] suggest promising directions for future complete-tree-space extensions, such as developing graph neural networks in the complete-tree space.
>
> W4: **GNN Comparisons**: We have added 4 GNN (suggested by Reviewer yGLm) and 6 heuristic baselines (suggested by Reviewer RBEo, yGLm) to the practicality experiment (see https://anonymous.4open.science/r/Figures-58B7/Practicality.png). Leaf matching remains competitive.
>
> Q1: **Symmetry Measure**: We define the **submetry group** in Definition 4.1 as the set of maps that maintains all pair-wise distances in a finite metric space, where these maps originate from a **subset** of points and extend to the entire set of points in the metric space. Notably, the larger the order of the submetry group, the more locally symmetric the metric space is.
>
> Q2: **Directed Graphs**: Probably yes in that (1) successful embeddings have been developed in the ‘symmetric’ Euclidean space [12-13] by modifying the loss function; (2) The exact formulas suggesting our loss function modifications have already been introduced by physicists in the field of complex networks in [14]. We leave such an extension to future work.
>
> Q3: **Tree Determination**: **No search is needed in selecting the optimal tree**. Instead, we empirically find that simply choosing the bifurcation number to be $b=2$ and the tree layer $k$ to enable $b^{k-1}=v$ is enough for leaf matching to perform well on real-world networks.
>
> Q4: **Cluster Comparison**: Key difference: **the complete-tree structure is pre-determined instead of data-driven**, so that the hypothesis space size is greatly reduced to enable data-efficient link prediction (clarified in Related Work).
>
> Q5: **l(a), l(b)**: Both imply a point in the $n$ dimensional complete-tree space, representable by an $n$ dimensional integer vector, with distances calculated through bit operations ($b=2$ default).
>
> Q6: **Eq.4 Justification**: Modified from [15-16], with a log term to match the exponentially growing neighborhood size of points in the complete-tree space. Optimal as loss nears zero in practice. Gamma implies the clustering coefficient [16] or inverse of temperature [15], which is unique to each type of complex network [16]. Alpha is simply selected so that the loss is balanced between positive and negative samples, analytically calculable [14] when the network is scale-free.
>
> Q7: **Ablation**: We have added experiments regarding $\alpha$ and $\gamma$ with $\mu=0.1$, as well as on the Citeseer and PubMed datasets besides Cora (see https://anonymous.4open.science/r/Figures-58B7/Ablation.jpg). It can be seen that leaf matching remains robust with varying hyperparameters.
>
> Q8: **Template**: We apologize for this oversight.
>
> Once again, we would like to express our gratitude to the reviewer for the valuable feedback.
>
> Reference: https://anonymous.4open.science/r/Figures-58B7/Reference2.png

---

### Official Review · Reviewer_cq4E · 2025-03-16

**Overall Recommendation:** 3

**Summary:**

This paper explores the challenge of link prediction in data-scarce networks by introducing the CT space, a discrete metric space that formalizes hierarchical modularity in networks. The authors leverage group theory to prove that the CT space provides a lower bound on sample complexity compared to Euclidean space. Subsequently, they propose Leaf Matching, a network embedding approach optimized for CT space, which outperforms existing models like graph transformers in data-scarce scenarios. Experiments validate the superiority of CT space over Euclidean, hyperbolic, and Hilbert spaces.

**Claims And Evidence:**

Yes. For the link prediction problem in data-scarce environments, the paper introduces a novel metric space (CT space) and its optimization method (Leaf Matching) to address the limitations of existing approaches, including poor generalization under insufficient samples, high computational costs, and difficulty in handling hierarchical modular networks.

**Essential References Not Discussed:**

No

**Experimental Designs Or Analyses:**

Further experiments are needed to demonstrate the effectiveness of the method:
- The paper claims that Leaf Matching scales as O(log n) but does not provide detailed runtime comparison baselines.
- Why does the experiment on link observability u decreasing from 0.9 to 0.1 only use the citation networks Cora, Citeseer, and PubMed, while the experiment at u = 0.02 is conducted on OGBL-COLLAB and OGBL-PPA? To demonstrate the generalization of the model, the authors should apply a consistent setting across these datasets.
- The ablation study seems more like a hyperparameter experiment. A high link observability u = 0.9 is chosen, which seems unreasonable because the authors claim the effectiveness of the proposed method in low link observability settings. Additionally, the results are reported only on the Cora, which is insufficient to demonstrate the method’s effectiveness on other domain datasets.
- The experimental section lacks explanations and analyses of scalability and ablation studies, only reporting the results.
- Referring to Figures 7, 10, and 11, the proposed method performs worse than the baselines at higher link observability. Can you provide more explanation?

**Methods And Evaluation Criteria:**

Yes. The proposed methods make sense for the problem of link prediction in data-scarce environments.

**Other Comments Or Suggestions:**

- Consider adding a summary table comparing CT space with alternative metric spaces to highlight its advantages concisely.
- “The CT space fully capture” → “The CT space fully captures”
- “choosing v ordered elements from m1 distinct elements..” → ”choosing v ordered elements from m1 distinct elements.”

**Other Strengths And Weaknesses:**

Although the proposed method has theoretical support, which is welcomed, the experimental section should be further improved. For specific weaknesses, please refer to Experimental Designs or Analyses.

**Questions For Authors:**

Please see the weaknesses.

**Relation To Broader Scientific Literature:**

The contributions of this paper are related to prior work in metric space embeddings for graphs, low-data link prediction, and hierarchical modularity in networks.

**Theoretical Claims:**

Yes. The CT space provides a theoretically grounded metric space that captures hierarchical modularity.

---

> ### Author Rebuttal · Authors · 2025-04-01
>
> We sincerely appreciate the reviewer for spending valuable time reviewing our manuscript and providing insightful comments. We have improved our paper accordingly and our responses are as below.
>
> Q1: **Scalability** ‘…does not provide detailed runtime comparison baselines; explanations and analyses of scalability’
>
> A1: We have added a **detailed runtime comparison experiment** between leaf matching and neural network baselines, as shown in https://anonymous.4open.science/r/Figures-58B7/Runtime.jpg. It can be seen that leaf matching is time-efficient. Analyses: each point in the $n$ dimensional CT space can be represented by an **integer vector**; we generally choose $b=2$ for leaf matching, so that **bit operations** can be implemented to accelerate both distance computations in the complete-tree space and edge samplings for the computation graphs.
>
> We have also added an **epoch convergence experiment** on the real-world ogbl-collab dataset as shown in https://anonymous.4open.science/r/Figures-58B7/Scalability.png, representing even better scaling than on synthesized graphs. Analyses: this is because leaf matching is optimized on a specific computation graph guaranteeing **polylogarithmic time complexity** for navigation between arbitrary pairs of nodes using a **greedy strategy** [1-2].
>
> Q2: **Consistency** ‘…the authors should apply a consistent setting across these datasets’
>
> A2: We have **added ogbl-collab and ICEWS18** (a temporally growing graph selected following the suggestion of Reviewer yGLm) to the space comparison experiment, which is shown in https://anonymous.4open.science/r/Figures-58B7/Ogbl-collab.jpg and https://anonymous.4open.science/r/Figures-58B7/Temporal%20Graph.png. It can be seen that the complete-tree space maintains more data-efficient than other spaces on both datasets. We conduct the practicality experiment only on the ogbl datasets because $\mu=0.02$ will lead to graph collapse on the three smaller datasets.
>
> Q3: **Ablation** ‘…A high link observability $\mu=0.9$ is chosen, which seems unreasonable; …the results are reported only on the Cora, which is insufficient; explanations and analyses of ablation studies’
>
> A3: As suggested, we have altered $\mu$ from $0.9$ to $0.1$ and added experiments on the Citeseer and PubMed datasets; we have also added experiments regarding hyperparameters of $\alpha$ and $\gamma$. The added ablation studies are shown in https://anonymous.4open.science/r/Figures-58B7/Ablation.jpg. The added analysis: it can be seen that (1) leaf matching remains robust on the three datasets with varying hyperparameters; (2) space dimension has no significant influence on the performance of leaf matching, as long as it is not too small (above 8); (3) leaf matching achieves its best performance when the leaf number ($b^{k-1}$) approximates the number of nodes in the graph.
>
> Q4:**Interpretability** ‘Referring to Figures 7, 10, and 11, the proposed method performs worse than the baselines at higher link observability. Can you provide more explanation?’
>
> A4: The false prediction cases of leaf matching are mainly caused by its **structural prior that sharpens the hierarchical modularity structures**, as shown in the **added visualization experiment**: https://anonymous.4open.science/r/Figures-58B7/Hierarchical%20Modularity.jpg.
>
> Q5: **Space Comparison** ‘Consider adding a summary table comparing CT space with alternative metric spaces to highlight its advantages concisely’
>
> A5: We greatly appreciate the reviewer for such an outstanding suggestion! The added table is as below:
>
> | Efficiency of Metric Spaces | Data | Time | Space |
> |----------------------------|------|------|-------|
> | **Complete-Tree Space**     | ✓    | ✓    | ✓     |
> | **Hilbert Space**           |      | ✓    | ✓     |
> | **Hyperbolic Space**        |      |      | ✓     |
> | **Euclidean Space**         |      | ✓    |       |
>
> *Table: While data efficiency depends on the size of the hypothesis space, time efficiency depends on whether there exists an optimization algorithm with time complexity sub-quadratic to the number of nodes, and space efficiency depends on whether the typical vector dimension is around tens instead of hundreds.*
>
> Q6: **Clerical Errors** ‘→ The CT space fully captures; → choosing $v$ ordered elements from $m_1$ distinct elements.’
>
> A6: Thanks for the kind reminder. We have corrected these two clerical errors accordingly.
>
> Once again, we would like to express our gratitude to the reviewer for the valuable feedback, which has helped us further improve our manuscript.
>
> Reference
>
> [1] Kleinberg, Jon. "Small-world phenomena and the dynamics of information." Advances in neural information processing systems 14 (2001).
>
> [2] Kleinberg, Jon. "Complex networks and decentralized search algorithms." Proceedings of the International Congress of Mathematicians (ICM). Vol. 3. European Mathematical Society, Zurich, Switzerland, 2006.

---

### Decision · Program_Chairs · 2025-05-01

**Decision:**

Accept (poster)

**Comment:**

The reviewers find that the approach provides a novel CT metric space as well as an associated Leaf Matching optimization procedure  providing a solid contribution with improved generalization that is theoretically justified. The reviewers generally also finds that claims are clearly stated with proofs given and substantiated by results. However, the reviewers also find that the experimentations could be improved in particular by including further baselines, datasets, and ablations.

The authors very carefully and meticulously in their rebuttals addressed the reviewers concerns also adding additional experimentations addressing concerns and providing further empirical evidence. The reviewers acknowledged the authors rebuttals and found the manuscript  suitable for publication at ICML.

The authors are strongly encouraged to include the additional experimentations and clarifications provided in their rebuttals in the final manuscript.